



# Hybrid forecasting: using statistics and machine learning to integrate predictions from dynamical models

Louise J. Slater[1], Louise Arnal[2], Marie-Amelie Boucher[3], Annie Y.-Y. Chang[4,5], Simon Moulds[1], Conor Murphy[6], Grey Nearing[7], Guy Shalev[8], Chaopeng Shen[9], Linda Speight[1], Gabriele Villarini[10], Robert L. Wilby[11], Andrew Wood[12], and Massimiliano Zappa[4]

[1]School of Geography and the Environment, University of Oxford, Oxford, UK
[2]University of Saskatchewan, Centre for Hydrology, Canmore, Canada
[3]Université de Sherbrooke, Canada
[4]Swiss Federal Research Institute WSL, Birmensdorf, Switzerland
[5]ETH, Zurich, Switzerland
[6]Irish Climate Analysis and Research Units, Department of Geography, Maynooth University, Kildare, Ireland
[7]Google Research, Mountain View, CA, USA
[8]Google Research, Tel Aviv, Israel
[9]Civil and Environmental Engineering, The Pennsylvania State University, State College, PA 16801, USA
[10]IIHR–Hydroscience and Engineering, University of Iowa, Iowa, USA
[11]Geography and Environment, Loughborough University, Loughborough, UK
[12]National Center for Atmospheric Research, Climate and Global Dynamics, Boulder, CO, USA

**Correspondence:** Louise J. Slater (louise.slater@ouce.ox.ac.uk)

**Abstract.** Hybrid hydroclimatic forecasting systems employ data-driven (statistical or machine learning) methods to harness and integrate a broad variety of predictions from dynamical, physics-based models – such as numerical weather prediction, climate, land, hydrology and Earth system models – into a final prediction product. They are recognised as a promising way of enhancing prediction skill of meteorological and hydroclimatic variables and events, including rainfall, temperature, streamflow, floods, droughts, tropical cyclones, or atmospheric rivers. Hybrid forecasting methods are now receiving growing attention due to advances in weather and climate prediction systems at sub-seasonal to decadal scales, a better appreciation of the strengths of machine learning, plus expanding access to computational resources and methods. Such systems are attractive because they may avoid the need to run a computationally-expensive offline land model, can minimize the effect of biases that exist within dynamical outputs without explicit bias correction and downscaling, benefit from the strengths of machine learning models, and can learn from large datasets, while combining different sources of predictability with varying time-horizons. Here we review recent developments in hybrid hydroclimatic forecasting and outline key challenges and opportunities. These include obtaining physically-explainable results, assimilating human influences from novel data sources, integrating new ensemble techniques to improve predictive skill, creating seamless prediction schemes that merge short to long lead times, incorporating modelled initial land surface and ocean/ice conditions, acknowledging spatial variability in landscape and atmospheric forcing, and increasing the operational uptake of hybrid prediction schemes.



## 1 Introduction: Defining hybrid forecasting and prediction

This review draws together two different but overlapping bodies of literature and research communities: operational forecasting, which uses dynamical and mostly linear empirical methods for forecasting, and machine learning (ML), which has recently begun to demonstrate more complex and novel non-linear approaches in geophysical (earth system) modeling. Our central question is: How might hydroclimate predictability be enhanced by merging dynamical predictions from physics-based weather, climate and/or land/hydrological simulation models with data-driven models?

In the context of this review, hybrid prediction is a notable case of data-driven prediction that reflects the deliberate choice to combine dynamical and empirical components. More specifically, a hybrid forecast or prediction is the use of a data-driven method to harness and integrate a broad potential variety of raw predictions from dynamical models – e.g. numerical weather prediction (NWP) or earth system model (ESM) predictions – into a final prediction product. We use the term 'data-driven' to refer to all empirical, statistical and ML approaches, ranging from simple linear regression to deep learning (see Tables 1-2). There are a variety of different types of hybrid model structure, ranging from statistical-dynamical (driving a statistical or ML model with dynamical meteorological or climate model outputs), to serial (combining ML and hydrological models sequentially, e.g. using a data-driven model to post-process the output of a conceptual model), and parallel approaches (merging ML and another type of model, e.g. replacing a component of a conceptual hydrological model with a data-driven model) (Figure 1; Table 3). We do not provide a prescriptive definition of hybrid forecasting as it exists along a continuum, which may include a wide range of modeling and 'big data' type Earth Observation (EO) datasets. In our review, we focus on a broad range of hydrometeorological variables, with a particular emphasis on streamflow.

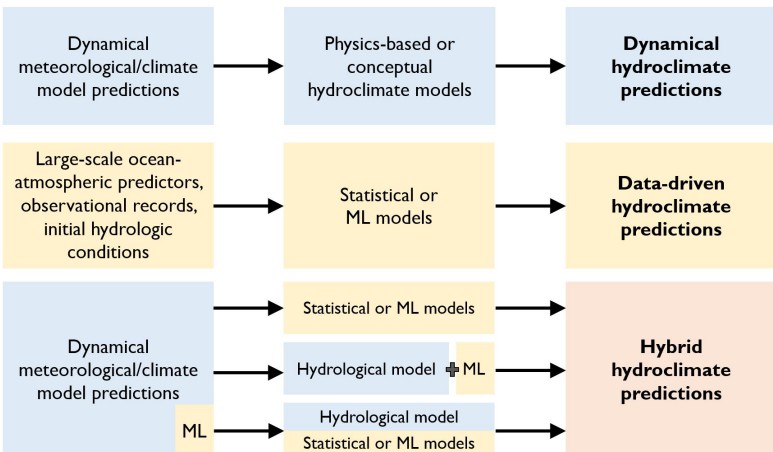

**Figure 1.** Defining hybrid forecasting and prediction in hydroclimatology. "Hydroclimate" here refers to streamflow and other hydrologic and climate variables defined in the text. Three broad types of hybrid model structure are shown in the three rows (from top to bottom): statistical-dynamical, serial, and a parallel model, which here additionally includes ML-based post-processing of dynamical model output (small insert in the bottom left box).



Traditional forecast workflows, in which a physics-based or conceptual land/hydrology model generates the final forecast
product, still make up the bulk of operational streamflow forecast systems worldwide. There is a long history of development
and application of standalone dynamical land surface and catchment hydrology models of varying complexity (from conceptual
to physically-explicit) for operational forecasting. Process-based hydrological modelling approaches may be physics-based or
conceptual, and gridded/spatially distributed or lumped. Examples include the hourly conceptual rainfall-runoff GR4H model
used by the Bureau of Meteorology in Australia (Hapuarachchi et al., 2022); the conceptual reservoir-based HSAMI model
implemented by Hydro-Québec (Bisson and Roberge, 1983); or the conceptual Sacramento Soil Moisture Accounting (SAC-
SMA) model of the Community Hydrologic Prediction System of the U.S. National Weather Service (Burnash et al., 1973).
In operational systems, the hydrological model is typically forced with outputs from NWPs, as in the case of the US National
Water Model (NOAA, 2016), the European and Global Flood Awareness Systems, EFAS (Thielen et al., 2009) and GloFAS
(Alfieri et al., 2013). (See Zappa et al. (2008) for a report on science-driven operational application of several end-to-end
ensemble hydrometeorological forecasting systems.) Outputs from coupled atmosphere-ocean-land general circulation models
(GCMs) may be used over longer time horizons, as is the case with the EFAS and GloFAS seasonal hydrological outlooks
(Smith et al., 2016; Arnal et al., 2018; Emerton et al., 2018; Harrigan et al., 2020). These approaches are often perceived by
users as more reliable in unseen conditions and physically interpretable than 'black box' statistical methods. However, the
large computational demand and variable skill of many traditional forecasting approaches still persists (Arnal et al., 2018).
While conceptual hydrological models may be less computationally expensive to run and more parsimonious than physics-
based models, their calibration still requires substantial effort (Arheimer et al., 2020; Hirpa et al., 2018) relative to many types
of purely statistical/empirical models. For instance, the calibration of the GloFAS system with an Evolutionary Algorithm
(EA) requires approximately 6 hours to calibrate each one of 1000s of streamflow stations on a 12-core PC, depending on
the number of generations needed before the improvement criterion is met (Hirpa et al., 2018). Training deep learning (DL)
models is orders of magnitude cheaper. For example, it takes about 10 hours to train an ensemble of Long Short-Term Memory
(LSTM) networks on a single NVIDIA V100 GPU using two decades of daily data from 518 basins in the CAMELS-GB
dataset (Lees et al., 2021), which works out to about 70 seconds per basin. This means that training a high-quality DL model
for hundreds of basins can be done on a standard workstation (or even a GPU-enabled laptop with sufficient memory), while
calibrating a conceptual or process-based model over hundreds of basins requires either months of runtime or an HPC facility.
In contrast with traditional forecast workflows, purely data-driven statistical or ML-based time-series prediction has histori-
cally relied more on observed data than dynamical climate model predictions, building empirical relationships between stream-
flow and precipitation and/or snow water equivalent (Garen, 1992), using time-lag relationships between upstream and down-
stream flow, or stochastic autoregression approaches like Auto-Regressive Moving Average (ARMA) models (Jain et al., 2018).
In such models, the hydrological predictands are regressed on a range of covariates, such as observed precipitation/temperature
records, static variables (e.g. elevation, slope, geology, land cover), initial hydrologic conditions, or large-scale predictors such
as sea surface temperatures (SST), surface air temperature, geopotential height, meridional wind, sea ice extent, or modes of
climate variability such as the El Niño-Southern Oscillation (ENSO) (e.g. Wilby et al., 2004; Dixon and Wilby, 2019; Mendoza
et al., 2017; Meißner et al., 2017). Methods such as canonical correlation analysis (CCA) (e.g. Barnston and Smith, 1996) can



be used to capture the linear relationships between predictor variables that explain the most variance in the predictand (Cohen
et al., 2019). Broadly speaking, the strength of statistical models lies in their simplicity, speed, ease-of-use, and comparable skill
to dynamical methods when there are sufficient observations for model training. Historically, statistical models were criticized
for their inability to predict extreme unforeseen conditions, since they often rely on stationary statistical relationships which
may not necessarily account for outlier values or shifts in the relationship between the predictand and predictors. Others have
raised the risk of artificial skill in statistical models where predictors are selected preferentially based on correlation with the
predictand (e.g. DelSole and Shukla, 2009). They may also be difficult to optimize for multi-variate, high-dimensional output
fields, which are simulated intrinsically by dynamical models. However, several recent studies have argued that deep learning
streamflow models can extrapolate to extreme unforeseen conditions (Frame et al., 2022a), to new catchments (Kratzert et al.,
2019a) (in cases where the catchments used for testing are from similar geographical regions to those used for calibration) and
to poorly gauged large regions (Feng et al., 2021; Ma et al., 2021) better than physics-based or calibrated conceptual hydrologic
models. However, there are some caveats here, as deep learning models still deteriorate substantially as they are extrapolated in
space. Thus, the outperformance over traditional models could be due to their decline from a higher baseline (in-sample basins)
rather than a stronger ability to generalize. There is still a significant amount of work that needs to be done to understand to
which types of basins and under what hydrological conditions DL models are able to extrapolate from the training set. Open
source modeling packages designed to make large sample hydrology applications more accessible and more computationally
efficient have been developed for Python (NeuralHydrology; Kratzert et al., 2022a) and R (Slater et al., 2019a).

Hybrid forecasting combines and draws on the strengths of these two different approaches and is growing in popularity
as a complementary tool alongside more traditional methods (Figure 1). Hybrid forecasts benefit from combining the ability
of physical models to predict and explain large-scale phenomena (i.e. through NWPs or climate model predictions) *with* the
strengths of statistical models to efficiently estimate probabilities or characteristics of events conditioned on observed data.
Many current examples of hybrid predictions build on the traditional forecast workflow by using an ML algorithm in sequence
with or alongside a conceptual or physics-based hydrological model (World Meteorological Organization, 2021) (Figure 1).
However, although hybrid prediction methods show considerable promise, they still largely fall outside the realm of traditional
forecasting and are rarely employed operationally (Cohen et al., 2019). This highlights how difficult it is to transfer scientific
innovation into decision making chains (Frick and Hegg, 2011).

The diversity of approaches for hybrid forecasting and prediction is evident from the range of techniques listed in Table 1.
The scope of hybrid models can vary widely, encompassing different forecast units (e.g. hourly or seasonal mean forecasts),
lead times (from the next hour to next decade, e.g. Ravuri et al., 2021; Neri et al., 2019), and geographical domains (from point
to street-level, single river catchment through to global approaches). Hybrid models have been applied to predict a variety
of hydrometeorological variables, including extreme heat and precipitation (Miller et al., 2021; Najafi et al., 2021; Miao
et al., 2019; Ma et al., 2022), seasonal climate variables (Golian et al., 2022; Baker et al., 2020), tropical cyclones/hurricanes
(Vecchi et al., 2011; Murakami et al., 2016; Kang and Elsner, 2020; Klotzbach et al., 2020), streamflow (Wood and Schaake,
2008; Mendoza et al., 2017; Rasouli et al., 2012; Duan et al., 2020), flooding (Slater and Villarini, 2018), drought (Madadgar
et al., 2016; Wu et al., 2021), sea level (Khouakhi et al., 2019), and reservoir levels (Tian et al., 2021), over a range of



**Table 1.** Examples of hybrid forecasts involving climate models of varying predictand, model type, and horizon (sorted by increasing time horizon). Acronyms are defined in Table 2. We also include some longer-term scenario-based projections for illustration purposes.

| Predictand | Statistical model | Dynamical model | Horizon | Citation |
|---|---|---|---|---|
| Daily streamflow | BNN, SVR, GP, MLR | NOAA GFS | 1-7 days | Rasouli et al. (2012) |
| Biweekly temperature and precipitation | PLSR | CFSv2 | 2–3 and 3–4 weeks | Baker et al. (2020) |
| Drought: seasonal SPI | D-Lasso & D-ANN | ECMWF SEAS5 | 1-90 days | Wu et al. (2021) |
| Seasonal tropical storm frequency | MLR | UKMO Glosea5 | 1 month | Kang and Elsner (2020) |
| Seasonal streamflow | PCR & CCA | CFSv2 & ECHAM4.5 | 1 month | Sahu et al. (2017) |
| Monthly reservoir inflow | RF, GBM, ELM, M5-cubist, elastic net | FLOR | 1 month | Tian et al. (2021) |
| Seasonal rainfall | ANN, MLR | UKMO GloSea5, ECMWF SEAS5 | 1-4 months | Golian et al. (2022) |
| Accumulated seasonal reservoir inflow | SVR, GP, LSTM, NLANN, DL | CMCC | 1-6 months | Essenfelder et al. (2020) |
| Discharge and surface water levels | MLR, LR, DT, RF, LSTM | ECMWF SEAS5; EFAS hydrological forecasts | 1-7 months | Hauswirth et al. (2022) |
| Hurricane frequency and intensity | GAMLSS | NMME (6 models) | 1-9 months | Villarini et al. (2019) |
| Seasonal runoff | PCR | NMME (7 models) & ECMWF SEAS4 | 4-9 months | Lehner et al. (2017) |
| Hurricane frequency | Statistical emulator of dynamical atmospheric model | GFDL–CM2.1 & NCEP–CFS | 1-10 months | Vecchi et al. (2011) |
| Seasonal streamflow | GAMLSS | NMME (8 models) | 1-10 months | Slater and Villarini (2018) |
| Monthly streamflow | FoGSS, CBaM | POAMA-M2.4 | 1-11 months | Bennett et al. (2016) |
| Seasonal flood counts | Poisson regression | 9/14 CMIP5 GCMs | 1-10 years | Neri et al. (2019) |
| Daily streamflow | TCNN (& others) | 4 GCMs from LOCA (CMIP5) | Decades (RCP8.5) | Duan et al. (2020) |
| Flood magnitude | LSTM (+5 GHMs) | 5 GCMs from ISIMIP-FT (CMIP5-6) | Decades (RCP8.5) | Liu et al. (2021) |
| Daily streamflow | DNN-PCE | 10 GCMs (CMIP5) | Decades (RCP8.5) | Zhang et al. (2022) |

timescales (Table 1). Certain other examples discussed in this review are not fully hybrid (e.g. ML models that are not driven
by NWM/ESM predictions) but serve to illustrate the possibilities of future hybrid systems. Hybrid hydroclimatic forecasts
and predictions have numerous operational and strategic applications, including water resources planning, reservoir inflow
management (Tian et al., 2021; Essenfelder et al., 2020), surface water flooding (Rözer et al., 2021), flood risk mitigation,
navigation (Meißner et al., 2017), and agricultural crop forecasting (Cao et al., 2022; Slater et al., 2021b). The envisaged
dynamical predictors may include various model outputs such as meteorological forecasts with lead times up to 14 days;
initialized climate predictions with sub-seasonal to decadal lead times; sub-seasonal runoff predictions, and/or land surface



**Table 2.** Modelling acronyms referred to in the manuscript. Top box includes data-driven models & approaches; bottom box includes other model acronyms

| Acronym | Full name |
| --- | --- |
| ANN | Artificial neural network |
| BAMLSS | Bayesian additive models for location, scale and shape |
| BMA | Bayesian model averaging |
| BNN | Bayesian neural network |
| CBaM | Calibration, bridging and merging |
| CCA | Canonical correlation analysis |
| D-ANN | Dynamic artificial neural network |
| D-Lasso | Dynamic lasso |
| DLNN | Deep-learning neural network |
| DNN-PCE | Deep neural network-based polynomial chaos expansion |
| DT | Decision tree |
| ELM | Extreme learning machine |
| FoGSS | Forecast guided stochastic scenarios |
| GAMLSS | Generalised additive models for location, scale and shape |
| GBM | Gradient boosting machine |
| GP | Gaussian process |
| LR | Lasso regression |
| LSTM | Long short-term memory |
| ML | Machine learning |
| MLR | Multiple linear regression |
| NLANN | Non-linear autoregressive neural network |
| PCR | Principal component regression |
| PLSR | Partial least squares regression |
| RF | Random forest |
| SVM | Support vector machine |
| SVR | Support vector regression |
| TCNN | Temporal convolutional neural network |
| GCM | Global climate model |
| GHM | Global hydrological model |
| CMIP5&6 | Coupled model intercomparison project phases 5 and 6 |
| ISIMIP | Inter-sectoral impact model intercomparison project |

or ocean state fields from the reanalyses used to initialize the climate system. Predictors are selected based on their ability to enhance hybrid forecast skill, such as traditional hydroclimate variables (e.g. precipitation, temperature, evapotranspiration) but also large-scale climate indices and teleconnections (e.g. DelSole and Shukla, 2009).

This paper provides an overview of recent developments and ongoing challenges in hybrid hydroclimatic forecasting. We
seek to highlight the benefits of employing hybrid methods alongside traditional forecasting systems based on physics-based or conceptual hydrological models. Accordingly, in Section 2, we provide several in-depth examples of different approaches to hybrid hydroclimatic forecasting. In Section 3, we discuss the key strengths of hybrid models, followed by ongoing challenges and future research opportunities in Section 4. We close with some concluding remarks in Section 5.



## 2 Hybrid forecasting

Hybrid forecasting encompasses approaches for pre-/post-processing hydroclimate predictions (Section 2.1), and for developing predictive models themselves, including short-term hybrid forecasts (Section 2.2), or sub-seasonal to decadal predictions (Section 2.3), and the integration of ML within parallel and coupled hybrid models (Section 2.4 and Table 3). The type of atmospheric and climate model outputs employed within hybrid models that span from single models to large multi-model ensembles. For example, there are the North American Multi-Model Ensemble (NMME, Kirtman et al., 2014) and the Copernicus

Climate Change Service (C3S) seasonal forecasting systems over subseasonal to seasonal timescales, or the Coupled Model Intercomparison Project (e.g. CMIP5-6) over decadal timescales. Many types of statistical model have been used (Tables 1-2), including simple regression methods (linear or quantile regression), principal components (Sahu et al., 2017), distributional regression frameworks such as the Generalized Additive Models for Location, Scale and Shape (GAMLSS), and various types of deep learning approaches, including ANNs and LSTMs.

**Table 3.** Examples of different hybrid model structures.

| Name | Description |
|---|---|
| Statistical-dynamical | Driving or conditioning a statistical or ML-based model with weather/climate/earth system model predictions - also called 'parameter informed' (e.g. Schlef et al., 2021; Slater and Villarini, 2018; Donegan et al., 2021). |
| Serial | Using a data-driven model to pre/post-process the output of a conceptual or mechanistic model (e.g. Lee et al., 2002). |
| Parallel | Replacing a component of a conceptual hydrological model with a statistical or ML model (e.g. Xiong and O'connor, 2002; Okkan et al., 2021) or, for instance, employing a data-driven model to estimate the difference between mechanistic model predictions and operational data (residuals) and then using the information content of those residuals (Lee et al., 2002). |

## 2.1 Pre- and post-processing of hydroclimate predictions using data-driven approaches

Hybrid hydroclimatological forecasting and prediction models may include pre/post-processing of inputs and outputs at different stages of the predictive model. Pre-processing refers to various techniques for enhancing the signal of the data inputs, such as the dynamical climate simulations; post-processing refers to techniques for refining and correcting model outputs. Depending on the point of reference, the same technique can be either pre/post-processing. It is important to point out that

pre/post-processing is also used in traditional forecasting systems (e.g. by driving a hydrological model with pre-processed climate predictions). Thus, by itself, pre/post-processing is insufficient to define a model as "hybrid". The strength of hybrid approaches lies in their ability to incorporate such corrections directly within hybrid modelling frameworks.

Data-driven models are frequently used for downscaling low-resolution climate model simulations to reduce precipitation bias and make the outputs more skillful at the catchment scale. For instance, Generative Adversarial Networks (GANs) have

been used to spatially downscale precipitation forecasts (Harris et al., 2022; Pan et al., 2022) to capture complex joint distributions between precipitation and initial climate conditions from climate simulations. Linear and kernel regression can be used to enhance the skill of decadal CMIP5 precipitation predictions (Salvi et al., 2017a). Similarly, quantile-based bias correction and regression-based downscaling can enhance the skill of GCM outputs (Salvi et al., 2017b; Meresa et al., 2021). For instance,





Monhart et al. (2019) supplied pre-processed (both bias-correction and downscaling via quantile mapping) temperature and
precipitation inputs to a conceptual hydrological model and showed skill improvements for streamflow and snow water equiv-
alent forecasts at subseasonal scale in small snow-dominated mountainous catchments in Switzerland. Random Forest (RF)
models can be trained to map low-resolution climate model outputs including precipitation to high resolution values (Anderson
and Lucas, 2018). Regardless of the algorithm used, once the mapping from low-resolution to high-resolution values has been
learned, the ML model can be applied to a much larger number of low-resolution model simulations to produce an ensemble of
high-resolution outputs at a considerably lower computational cost than running a dynamical model at an equivalent resolution.
Another example is the use of data-driven methods to reduce the degrees of freedom in data, e.g. through discrete or empirical
wavelet transforms (Mosavi et al., 2018).

ML models are also employed during the dynamical climate model simulations to correct model biases (e.g. Schepen et al.,
2018; Watt-Meyer et al., 2021; Wang et al., 2021). For example, a RF model coupled to an atmospheric model (FV3GFS)
can correct temperature, specific humidity and horizontal winds at each timestep, bringing the coupled model in line with
observations. This was shown to reduce annual-mean precipitation biases by around 20%, with particular improvements in the
simulation of rainfall over high mountains (Watt-Meyer et al., 2021). A similar approach was used by Bretherton et al. (2022)
to nudge the output of a low-resolution climate model towards the coarsened output of a high-resolution climate model. The
coupled model showed improvements in surface latent heat flux and rainfall by correcting the insufficient cloud cover seen in
the low-resolution model output. It has to be also kept in mind that, if possible, multi-variate bias-correction should be preferred
to uni-variate approaches in impact modelling (Meyer et al., 2019; Wang et al., 2021).

The use of ML models for post-processing can also be applied directly to the hydrological forecasts. Bennett et al. (2021a)
deployed an ERRIS (error reduction and representation in stages) error model to directly correct errors in streamflow prediction
up to 168 hours ahead (i.e., maximum lead time of 7 days). This approach can be especially beneficial for longer forecast
horizons. For instance, a Gaussian Process (GP) model was trained to post-process weekly tercile forecasts of runoff and soil
moisture from a Swiss conceptual hydrological model PREVAH, and showed improvements in the forecast skill up to 4 weeks
ahead (Bogner et al., 2022). McInerney et al. (2022) developed a daily Multi-Temporal Hydrological Residual Error (MuTHRE)
statistical model to seamlessly transform daily streamflow forecasts to time scales ranging from daily, weekly, fortnightly to
monthly. This one-model-for-all-scales approach is a novel take on the potential of the hybrid forecasting system.

Data-driven models can enhance the signal of predictors by pooling (generating an ensemble of) different climate model
predictions (Troin et al., 2021). A common approach to incorporate an ensemble of climate model predictions (within a statis-
tical, ML, or hydrological model, for instance) is to assume that predictions from each ensemble member are equally likely.
However, owing to varying model skill, as well as a lack of independence amongst some models, the assumption of equal
likelihood can be compromised. Hence, hybrid forecasting can be used to combine ensembles in more intelligent ways by
accounting for the varying information content of ensemble members. Statistical ensembling/post-processing of climate model
ensemble outputs can improve forecast skill at relatively low computational cost. For instance, Grönquist et al. (2021) applied
a deep neural network to ensemble predictions to improve forecast skill and reduce the computational requirements of the
forecast system. Massoud et al. (2020) applied Bayesian Model Averaging (BMA) to weight models according to their skill





at reproducing observations. They show the weighted ensemble average skill for the contiguous Unites States exceeds that
of the conventional ensemble average, with better constrained uncertainty estimates. Bayesian updating can also be applied
to enhance the skill of a multi-model ensemble of GCMs such as the NMME for different seasons or lead times (e.g. Slater
et al., 2017). Bayesian updating provides the best results when the raw GCM predictions have high skill to begin with, such
as SST-based ENSO forecasts (Zhang et al., 2017). Post-processing hydrological forecasts (instead of climate forecasts) is
another application of BMA. Hemri et al. (2013) demonstrated how such an approach can be deployed to improve the skill of a
conceptual runoff forecast by pooling four separate runoff forecasts forced with different lead times (24-hr, 72-hr, 120-hr, and
240-hr) and ensemble members (1, 1, 16, and 51, respectively) in a Swiss catchment.

Hybrid methods may also exploit large model ensembles to capture non-linear interactions between predictor variables. Such
approaches are valuable in cases where the complexity of the interactions is such that the number of observations required far
exceeds the available observational record (of around 40-100 years). For instance, Gibson et al. (2021) trained ML models
for precipitation forecasts in the western United States (driven by observed atmospheric and oceanic states prior to the target
season) on a large historical climate model ensemble (i.e. on thousands of seasons of simulations from the Community Earth
System Model Large Ensemble, CESM-LENS). The resulting ML-based approach performed as well as, if not better than,
seasonal NMME forecasts, and the physical processes could be interpreted using ML interpretability plots (highlighting the
most important variables that influenced a given forecast).

## 2.2 Short term hybrid forecasts (hours to weeks)

Short-term hybrid forecasts focus on outlook horizons of hours to weeks driven by dynamical meteorological models. Existing
physics-based and conceptual models are well-established in operational forecasting at this time scale and the skill of such
models is strongly influenced by the skill of the meteorological forecast. Consequently, the development of hybrid approaches
is less mature than at the seasonal scale, and we focus less on this horizon herein. Where hybrid approaches offer immediate
potential is to address the challenge of forecasting floods from convective rainfall with short lead times (Speight et al., 2021).
In these situations, the time taken to transfer data between meteorological and hydrological organisations, and the run time
of physics-based models can be restrictive. The potential of ML as a means to post-process dynamical forecasts and produce
warning scenarios for convective weather is emerging (e.g. Moon et al., 2019; Flora et al., 2021) but has not yet been widely
utilised as input to hydrological models. Like all models at this scale, the development of hybrid approaches is constrained by
the lack of observational data at the spatial and temporal scales required. Nonetheless, in their review of real time forecasting
in urban drainage systems, Piadeh et al. (2022) found that 9% of published models used hybrid approaches, where ML was
implemented either in a hybrid approach by combining physical and data driven models, or by using dynamic ML models
(e.g. LSTMs used by Zhang et al., 2018) to increase the available lead time. In this context, the strengths of ML are the small
number of input parameters making the models easy to develop, quick to run, and accurate over short lead-time events (Piadeh
et al., 2022).

At 1-7 day lead times, Rasouli et al. (2012) find that ML models outperform MLR (Tables 1-2) and that their hybrid approach
works best when driven by observations and the NOAA Global Forecasting System (GFS) model output at shorter lead times,





and by a combination of local observations and climate indices at longer lead times. At the days-to-weeks time frame, ML

offers promising alternatives to physics-based hydrological and inundation models for flood forecasting, particularly in regions
where access to local forecasting models is limited (e.g. Nevo et al., 2022), and shows potential to overcome limitations of
data scarcity (Kratzert et al., 2019a; Feng et al., 2021). For hydrologic forecasts, ML is highly successful in assimilating recent
observations of streamflow to improve near-term daily forecasts of streamflow (Feng et al., 2020). Although Google's ML flood
warning system was used successfully during the Indian and Bangladesh monsoons in 2021 (Nevo et al., 2022), the expansion
of such approaches to extend lead time for decision makers by employing forcing from dynamical meteorological forecasts

remains under explored.

## 2.3 Sub-seasonal to decadal hybrid forecasts

Combining dynamical climate predictions with ML models is still an emerging field of research. The vast majority of ML
predictions in the literature are driven by observational or reanalysis atmospheric data; they are not designed for an operational
forecasting context. In contrast, the majority of hybrid schemes are based on climate model outputs used to drive statistical

models, as described below.

A simple example of a hybrid statistical-dynamical model is one that employs the predictions of precipitation or temperature
from a climate model as predictors (covariates) within a regression model, where the predictand can be a hydroclimatic variable
such as streamflow magnitude (e.g. Slater et al., 2019b) or flood duration (Neri et al., 2020). Schlef et al. (2021) describe
this approach as an 'informed-parameter approach' in which the parameters of the flood distribution can be conditioned on

time-varying covariates such as time, climate indices, infrastructure development indices, or land use indices. For example,
distributional regression models can be used to predict seasonal discharge. To illustrate the approach, we consider a 9000 km$^2$
catchment that has experienced rapid expansion of the agricultural land area over the 20$^{th}$ century (Figure 2). Two lumped
(catchment-averaged) covariates are employed to predict the seasonal maximum of mean daily streamflow ($Q_{1.00}$) in each
year: the basin-averaged total seasonal precipitation ($x_r$) and the harvested corn and soybean acreage ($x_a$) in the same season.

The model employs a gamma distribution which has two parameters, $\mu$ and $\sigma$, with an expected value of streamflow ($Y$) equal
to $\mu$ and variance equal to the product of $\mu^2$ and $\sigma^2$. The variability of these two parameters can be described over time in
terms of the covariates $x_r$ and $x_a$. As the regression is probabilistic, the entire streamflow distribution is computed for each
timestep (the 5th-95th percentiles of the distribution are shown as a yellow ribbon in Figure 2). Having trained the model over
the historical period, the framework can be employed for probabilistic forecasting. Model parameters are extracted for a given

historical period, and the forecast is based on those parameters and the predictions of the covariates. Once new observations
become available, the model can be retrained, updating the model parameters. A different model can be developed for each
season, initialization time (e.g. 0.5, 5.5 and 9.5 months ahead of a given season), and quantile of the predicted discharge
distribution.

Seasonal forecasts of diverse hydroclimatic variables such as precipitation, evaporation, sea water level, sea level pres-

sure or large-scale climate indices have also been used to drive ML models of precipitation (Madadgar et al., 2016), stream-
flow, and tropical cyclone activity (Sabeerali et al., 2022). For instance, atmosphere-ocean teleconnections obtained from the

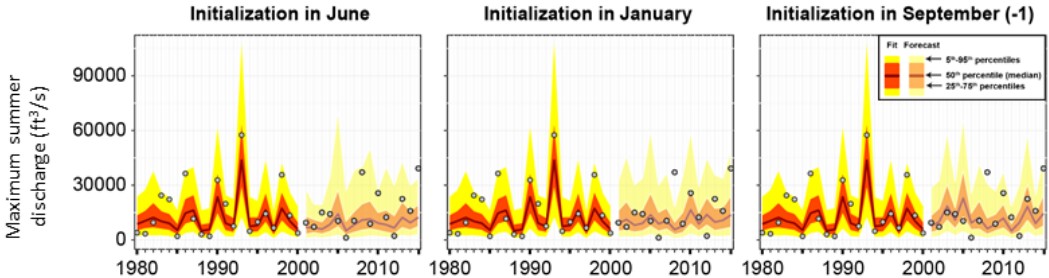

**Figure 2.** Example of seasonal hybrid forecasting system for maximum summer discharge at one stream gauge, using seasonal climate forecasts from 8 climate models (94 members) of the NMME to drive a distributional regression model of streamflow. The time series shows the model fit (1980-2000) and forecast (2001-2015) against the observations of maximum summer daily streamflow (grey circles). Initialization times are 0.5, 5.5 and 9.5 months ahead of the summer season. For example, "initialization in June" includes forecasts for 0.5-month lead for June, 1.5-month lead for July and 2.5-month lead for August (to compute the summer streamflow), while "initialization in September (-1)" includes forecasts initialized 9.5, 10.5 and 11.5 months ahead in the previous year. Skill could potentially be increased by using ML instead of GAMLSS and weighting the contribution of different model members according to their skill for different lead times, streamflow quantiles, or seasons. Modified from Slater et al. (2019b).

NMME – including the Pacific Decadal Oscillation (PDO), Multivariate ENSO Index (MEI), and Atlantic Multidecadal Oscillation (AMO) – were used to successfully predict seasonal precipitation anomalies using a statistical Bayesian-based model (Madadgar et al., 2016). For Ireland, Golian et al. (2022) found that MLR and ANN models applied to hindcasts of mean
sea level pressure from GloSea5 and SEAS5 produced skillful forecasts of winter [DJF] and summer [JJA] precipitation for lead times of up to four months, with the ANN outperforming MLR for both seasons and all lead times. A study over the Netherlands using streamflow, precipitation, and evaporation found that the hybrid ML approach outperformed climatological reference forecasts by approximately 60% and 80% for streamflow and surface water level, respectively, using various machine learning models (Hauswirth et al., 2022). Another study employed predictions of large-scale indices by the CFSv2 model to
predict the frequency of tropical cyclones in the Bay of Bengal using principal component regression (Sabeerali et al., 2022).

Hybrid forecasts have also been tested for subseasonal to seasonal impact applications such as energy forecasting. In southern Switzerland, for instance, five ML models were trained to predict monthly total hydropower production, by combining precipitation, temperature, radiation, and windspeed forecasts from a dynamical meteorological model with runoff from a conceptual hydrological model (Bogner et al., 2019). Day of the week and holiday information were provided to the ML models
as additional information to further enhance the prediction.

Statistical-dynamical approaches can also be deployed for much longer horizons, such as decadal streamflow predictions (e.g. Neri et al., 2019). Different techniques exist to enhance the skill of hybrid streamflow forecasts, such as training the model directly on the climate model predictions instead of observations (e.g. Slater and Villarini, 2018), or via novel techniques such as "mode-matching" (Moulds et al., 2022). Large climate ensembles can be pre-processed to select members which are skilful





at a given time, and the improved predictions can then be supplied to a statistical modelling framework to predict seasonal streamflow quantiles (Moulds et al., 2022).

## 2.4 Hybrid forecasts including a conceptual hydrological model

Conceptual and data-driven hydrological models have been compared in the scientific literature. Some studies indicate that conceptual models are less prone to overfitting than data-driven models (e.g. Jeong and Kim, 2005); others assert that data-driven models generally outperform conceptual models in terms of the final result, such as forecasted streamflow (e.g. Daliakopoulos and Tsanis, 2016). Hybrid forecasting systems based on conceptual hydrological models try to combine strengths rather than compete. The well-known conceptual rainfall-runoff model GR4J (Perrin et al., 2003) has been used extensively to construct hybrid models. For instance, Humphrey et al. (2016) used simulated soil moisture from GR4J, intended to represent initial conditions, to a Bayesian neural network to forecast 1-month ahead streamflow for three catchments in southern Australia. They showed that the hybrid model is not only significantly more precise than climatology, but also outperforms both GR4J alone and a Bayesian neural network alone. Both Anctil et al. (2004) and Kumanlioglu and Fistikoglu (2019) replaced the routing component of the GR4J model with an ANN, for catchments in France, the USA and Turkey. These studies concluded that the hybrid model was superior to a purely ML model. Similarly, Kumanlioglu and Fistikoglu (2019) compared a hybrid model with GR4J and found the former performed best. This type of model can be described as *parallel hybridisation* (Okkan et al., 2021).

Other conceptual hydrological models have been used in hybrid frameworks. For example, Mohammadi et al. (2021) used two conceptual models, HBV (Bergström, 1976) and NRECA (Crawford and Thurin, 1981) to provide inputs to support vector machines (SVM) and adaptive neuro-fuzzy inference system (ANFIS), to build seven variants of hybrid models. They tested and compared the hybrids as well as the individual models (HBV, NRECA, SVM and ANFIS) on four sub-basins of the Pemali Comal River Basin, Indonesia, and again found the hybrid models performed best in terms of RMSE, $R^2$ and MAE. Such hybrid models, where the data-driven model is used to post-process the output of the conceptual model, are referred to as *coupled models* (Okkan et al., 2021). Other studies on hybrid modeling using the HBV model include Nilsson et al. (2006) and Ren et al. (2018). They both used different variables computed by HBV (e.g. soil moisture, snowmelt) as inputs to ANNs.

Okkan et al. (2021) outline that in most hybrid modeling frameworks, variables computed by the conceptual model are used as inputs to a data-driven model, which necessarily increases computation time. They also note that although there could potentially be interactions between the parameters of the conceptual models and those of the data-driven model, those interactions often go unaccounted for because the two models are calibrated separately. In the context of monthly rainfall-runoff modelling, they proposed to address these two common shortcomings of hybrid models by coupling the two models and performing their calibration jointly.



## 3 Strengths of hybrid forecasting

Hybrid methods offer various strengths, as summarized in Figure 3. These include benefits related to the higher performance of ML models (in terms of bias minimisation and greater accuracy), the ability to easily blend outputs from climate multi-model ensembles, integrating large datasets, combining multiple sources of predictability to enhance predictive skill, improved speed and operational convenience. These strengths are discussed in more detail below.

### 3.1 ML model performance and bias minimization

Recent work has demonstrated the ability of ML models to outperform traditional hydrological models (e.g. Kratzert et al., 2019b; Lees et al., 2021; Feng et al., 2020; Fang and Shen, 2020; Fang et al., 2017). Mai et al. (2022) compared 13 locally- and globally-calibrated models (including ML, lumped and gridded models) in terms of their ability to simulate streamflow, actual evapotranspiration, surface soil moisture and snow water equivalent in the Great Lakes region. They found that the ML model outperformed the traditional hydrological models in all experiments. This finding extends to ungauged catchments: Kratzert et al. (2019a) found an out-of-sample LSTM performed better than either the calibrated SAC-SMA (the conceptual model under the US CHPS system) or the National Water Model. Golian et al. (2021) found that random forests worked best at regionalizing the parameters of the GR6J conceptual model for low flow prediction in ungauged Irish catchments. Such work has shown the potential of hybrid methods to address the longstanding hydrological challenge of prediction in ungauged basins (e.g. Sivapalan, 2003). The next step is moving from modelling to prediction.

Hybrid models combining ML and climate predictions also tend to outperform the raw dynamical forecasts from climate models. Wu et al. (2021), for instance, developed a hybrid drought-forecasting model of the 3-month Standardised Precipitation Index (SPI). They employed ECMWF SEAS5 predictions of geopotential height, sea level pressure and air temperature to force two ML models, dynamic-Lasso and dynamic-ANN. They found that the SPI predictions from these hybrid models outperformed the predictions of SPI obtained directly from Meteo France, ECMWF and UKMO model outputs. For prediction purposes, hybrid models have the advantage of being able to minimize biases that exist within GCM outputs or that might be otherwise introduced within a hydrological modelling chain. By training a hybrid model directly on the climate model forecasts/predictions, rather than on observations, the biases are automatically accounted for within the model (e.g. Slater and Villarini, 2018). This approach is similar to that of model output statistics (MOS) long used by the weather forecasting community (Glahn and Lowry, 1972), but also in seasonal hydrological predictions (Schick et al., 2018). For instance, if a climate model tends to overpredict winter rainfall, this bias is accounted for directly in the streamflow predictions, given that the model is trained using the same winter rainfall forecasts (assuming a constant bias).

Hybrid models may benefit from a wide range of statistical advances for enhancing the skill of hydroclimate predictions. Since a hybrid system is based on a statistical model, it is straightforward to incorporate statistical 'upgrades', such as ensembling the outputs of multiple climate/earth system models (Duan et al., 2019). One such example is the addition of an error model onto Ensemble Streamflow Prediction (ESP) forecasts to enable prediction in ephemeral rivers (Bennett et al., 2021b). In a hybrid system, one may easily integrate the predictions from multi-model ensembles with over 50 or 100 model members





as covariates (Gibson et al., 2021; Slater and Villarini, 2018). Increasing the number and diversity of climate models included within a hydrological predictive model enhances confidence in the hydrological model spread. By blending multimodel ensem-

bles intelligently one can further reduce uncertainty. In a hybrid system, for instance, one can incorporate time-varying weights for the dynamical predictions, such as Bayesian updating - varying model weight per month and lead time (Slater et al., 2017). ML models especially can learn space-time variable input weighting directly (Kratzert et al., 2021). Similarly, many post-processing methods can be applied to weather and climate inputs or the hydrological outputs to enhance skill (Monhart et al., 2019; Bogner et al., 2022).

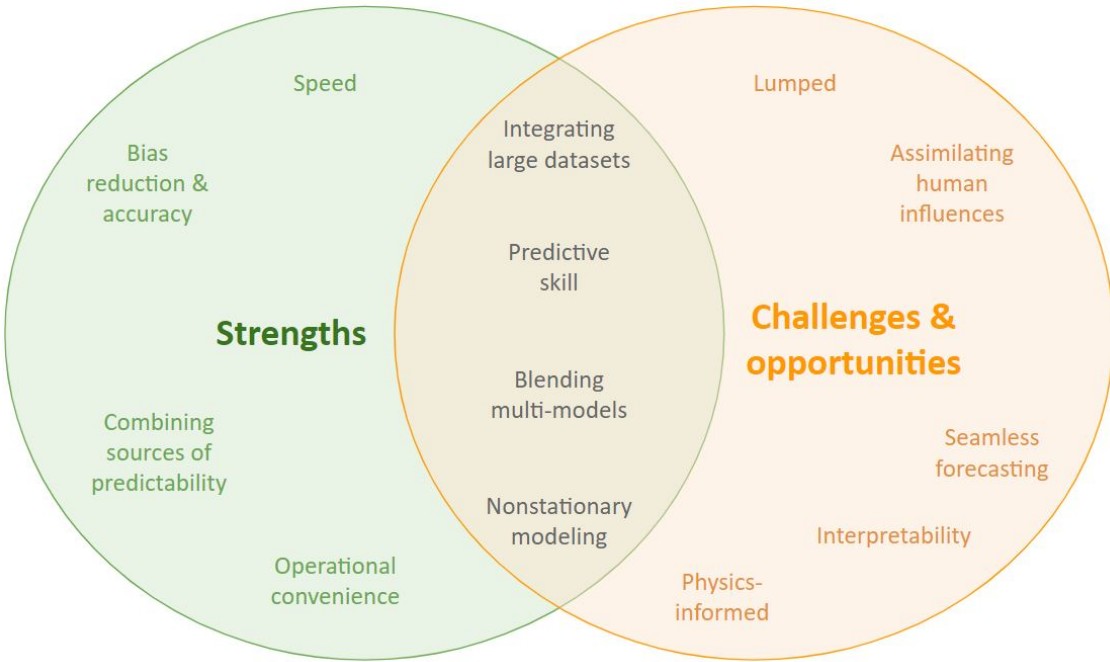

**Figure 3.** Strengths, challenges and opportunities of hybrid hydroclimate prediction systems, as discussed in Sections 3 and 4.

## 3.2    Combining different sources of predictability with varying time-horizons

One under-researched but promising aspect of hybrid models is their ability to combine different sources of predictability over a continuum of time horizons. Hybrid models can easily make use of different predictors chosen on a sound physical basis (such as climate indices, precipitation, air pressure, snowfall) without explicitly describing the processes and equations. This makes it much easier to explore information from new sources and improve models, and has the potential to widen information

access to climate-affected populations. Including additional inputs can also produce marked improvements in model quality. Chang et al. (2022, under review) used seven weather regime indices (based on the 500 hPa geopotential height) with a Gaussian Process ML model to post-process sub-seasonal hydrological forecasts, alongside runoff, soil moisture, baseflow, and snowmelt in Switzerland. The results showed that the additional input of weather regime indices improved the forecast



skill especially in the mountainous catchments and over longer lead times, where skill was difficult to improve without any
additional information. The conceptual hydrological model would not have been able to take weather regime indices as input,
but by including them in the post-processing ML model as part of the hybrid setup, it was possible to explore the connection
between large scale weather regimes and local hydrological conditions to improve the forecast skill.

As multiple predictor variables can be included within a statistical or ML model, it is feasible to combine predictors that
have very different time-varying impacts, such as reservoir management decisions or initial hydrological conditions impacting
the short term, versus annual-to-multidecadal climate oscillations for longer-term predictability. For instance, Tian et al. (2021)
present a reservoir inflow forecasting framework combining a suite of different ML models (including gradient boosting ma-
chine, random forests, and elastic net) with climate model outputs from the FLOR model, for reservoirs in the Upper Colorado
River Basin. They also included soil moisture and evaporation to represent antecedent conditions, which significantly improved
the forecasts of reservoir inflow. Ouyang et al. (2021) used a dataset of >3000 basins across the USA and found that basins
with small and medium reservoirs behave characteristically differently from reference basins, but can be well simulated by a
LSTM model with input attributes describing basin-lumped reservoir statistics.

Large-scale climate indices or modes are often combined with other predictors. For instance, Madadgar et al. (2016) pre-
dicted seasonal precipitation using large-scale climate indices: the PDO, the MEI, and the AMO, computed from outputs of the
99 ensemble members of the NMME. The approach enhanced the skill of the seasonal forecasts by 5-60% in comparison with
the raw NMME precipitation forecasts, especially for negative rainfall anomalies. Similarly, Rasouli et al. (2012) forecasted
daily streamflow in a river catchment 1-7 days ahead by employing weather forecasts from the NOAA GFS model within a
variety of machine learning models. They combined observations with the model outputs and also a range of large-scale cli-
mate indices representing ENSO, the Pacific-North American teleconnection (PNA), the Arctic Oscillation (AO) and the North
Atlantic Oscillation (NAO). Lastly, Li et al. (2022) used forecasts of the intraseasonal oscillation (ISO), an important mode
of subseasonal predictability for seasonal rainfall, to force a Bayesian hierarchical model predicting sub-seasonal precipitation
during the boreal summer monsoon season in different regions of China.

Given the diversity of potential inputs to hybrid forecasting systems, exploratory data analysis to identify correlations be-
tween hydrologic variables and climate patterns over different time horizons is an important step during model development.
For instance, Hagen et al. (2021) employed ML to identify the most relevant large-scale climate indices for daily streamflow
forecasting. They provided an overview of studies that have employed large-scale climate indices and climate variables (such as
sea level pressure, sea surface temperature, specific and relative humidity) within ML models for daily, monthly and seasonal
streamflow modelling. Beyond the use of pre-defined climate indices, it is possible to identify tailored, site-specific climate
indices from big data and incorporate them in the modelling chain. For instance, Renard and Thyer (2019) described a method
that avoids relying on standard climate indices and instead suggests that the most relevant climate indices in a given location
are effectively unknown (they are 'hidden') and can be estimated directly from observations. The authors used a Bayesian
hierarchical model for flood occurrence, with hidden climate indices treated as latent variables. They identified hidden climate
indices and then showed their correlation with atmospheric climate variables (geopotential height, zonal westerly wind, but
also more distant teleconnections using convective available potential energy and meridional wind). These indices explain the





occurrence of flood-rich and flood-poor periods in the historical record. Such an approach could be employed using climate
model outputs to develop skillful hybrid forecasts.

Related to the different time-horizons of the predictors is also the ability to design hybrid forecasting systems which dynamically update when new information (e.g. observations or climate hindcasts) become available. For instance, a statistical model can be updated iteratively over time to track the evolution of nonstationary predictor-predictand relationships. Such approaches incorporate new observations as they become available and update the model parameters (e.g. Slater et al., 2019b). Nearing
et al. (2021a) developed a data assimilation approach for LSTM models that leverages tensor network gradients to assimilate real-time observation data. To date, very little has been published using such methods.

### 3.3   Integrating large datasets

One perceived challenge of hybrid approaches is the requirement for large amounts of training data to constrain models compared with physics-based or conceptual models. Previously, it was felt that the information requirement of data-driven ap-
proaches might hinder their applicability in catchments with limited data (e.g. ungauged basins). Although this might have been true in the past, the increasing availability of large-scale hydrology datasets (e.g. remote sensing data) is turning this potential challenge into a new opportunity for ML. An ML model can be trained on the same data as a conceptual model, and will usually out-perform physics-based models, on average (and even more so with large datasets; see Fang et al., 2022). Large training datasets tend to be useful for ML but not so much for physics-based models as their ability to adapt would saturate;
the ability to leverage large datasets effectively is a strength of ML, and in particular for ungauged basins, where ML models tend to have higher accuracy, on average, than physics-based models calibrated in gauged basins (Kratzert et al., 2019a). There is, in fact, a 'data synergy' effect where more data and more diverse data lead to better models, according to a systematic study of LSTM models for either streamflow or soil moisture (Fang et al., 2022). With conceptual and process models, accuracy is typically lost when performing regional (as opposed to basin-specific) calibration, and the lack of calibration data typically
results in predictions with poor quality (training on longer periods leads to superior results – see Bogner et al. (2022)). In contrast, with hybrid models, strong performance can be achieved when training the models on global datasets, and accuracy is gained when performing regional calibration.

Since long (50-year +) hydroclimatic time series data are not available everywhere (Krabbenhoft, 2022), methods are required that draw on pooled multi-site approaches with similar catchment and climate characteristics (Kratzert et al., 2019a).
For instance, Nearing et al. (2021b) show a comparison using pooled vs unpooled data for streamflow estimation and found the former was better, even for gauged catchments, and allowed for prediction in ungauged catchments. There are, however, few studies combining LSTM methods with climate model outputs for long-term (subseasonal to decadal) prediction, especially in ungauged catchments. Such models may start to emerge with the growing availability of observational training datasets, such as the national 'CAMELS' and international 'Caravan' streamflow datasets (Kratzert et al., 2022b). However, real-time data
are currently still difficult to access for developing predictive models.

One way to circumvent the lack of observational training data and the low predictability of GCMs is by integrating a range of other types of predictors in hybrid approaches. This may include sources of remotely sensed measurements such as snow,





soil moisture, land cover, surface water extent, water storage or evapotranspiration to provide better information about initial states (e.g. Jörg-Hess et al., 2015). There are many different global datasets now available that can be drawn on. For example,

this can be done by employing Google Earth Engine to extract data directly, as was the case for the creation of an open-source community dataset for streamflow (Kratzert et al., 2022b).

The forecasting landscape is becoming increasingly complex, with a growing number of forecasting systems and datasets potentially overwhelming users. Hybrid forecasting could help to address this challenge, with hybrid workflows providing a set of tools and data that forecasters could mix and match to address their own forecasting needs.

### 3.4   Speed and operational convenience

A key advantage of statistical or hybrid methods is their speed and computational efficiency. As previously discussed, the training time for statistical or hybrid models can be a matter of seconds, or up to several days in the case of more complex models, depending on the computing power, number of locations and amount of data involved, compiler, and optimization. While deep learning methods such as LSTMs can take several hours to train (e.g. Lees et al., 2021), they have the significant

advantage that one model is trained on multiple sites (although the fitted model can then be fine-tuned to a specific site). A differentiable (machine-learning-based) parameter learning scheme can be trained on satellite-based soil moisture observations for the entire continental United States with one GPU in under one hour, but the conventional approach would take a cluster machine of 100 CPUs 2-to-3 days to calibrate the model (Tsai et al., 2021).

This efficiency has advantages for water managers. In a traditional setting with limited computational resources, to make the

best use of ensemble forecasts water managers need to quickly run different scenarios (Scher et al., 2021). For instance, the UK Flood Forecasting Centre will produce a "reasonable worst case" and a "best estimate" (most likely scenario; see Met Office, Environment Agency and Flood Forecasting Centre (2013)) ahead of a flood event (Arnal et al., 2020). Using all available deterministic and ensemble forecast products alongside expert assessment from the chief forecaster they will decide what the reasonable worst case is likely to be. These outputs are used to inform the flood guidance statement and the Environment

Agency uses these scenarios to run their catchment models (Pilling et al., 2016). The speed of data-driven approaches in comparison with these more traditional physics-based modelling approaches could prove beneficial for users wishing to run various scenarios quickly. Hybrid methods may shorten the traditional forecasting approach by going 'end-to-end', potentially skipping out some of the intermediary steps in a conventional modelling chain, such as downscaling, bias correction and hydrological modelling. This offers significant potential for applications where the run time of physically based models limits

the ability to provide forecasts with a useful lead time for action – such as surface water (Rözer et al., 2021) or flash flood forecasts.

The efficiency of hybrid models may also be helpful in generating faster research cycles for model improvements (i.e. setting up an upgraded system and releasing hindcasts for testing) relative to traditional approaches. Model upgrades for dynamical systems usually take a very long time because the model has to be re-calibrated and a set of $X$ years of hindcast data must be

produced for verifying and demonstrating the impact of the changes to the system.





Lastly, hybrid systems can be used to develop customized climate services. For instance Essenfelder et al. (2020) use data-driven methods to predict seasonal reservoir inflows for hydropower plants. The information is made easily accessible online to support decision-makers in hydropower production. Such approaches can be designed to be replicated globally as a climate service, provided there are suitable data for training, and by developing transferable rule sets. Bennett et al. (2016, p.8239)

also highlight the importance of operational convenience and the advantages of combining "the convenience of stochastic scenarios with the skill of a modern forecasting system". Their method enhances precipitation forecasts necessary for streamflow forecasting through post-processing - by reducing the biases, correcting the reliability, and maximising the forecast signal.

## 4 Key challenges and opportunities of hybrid forecasting

Beside the strengths of hybrid methods, what are some of the challenges and research priorities to be tackled? As hybrid

forecasts and predictions rely on statistical and ML models, they inevitably inherit some of the limitations of these techniques. Frequently-cited limitations of ML models include the requirement for large datasets (previously discussed) and the difficulty of obtaining physically plausible results for previously 'unseen' extremes. However, new research suggests that ML models may even provide results that are more physically plausible than physics-based and conceptual models – for instance, when data are biased (Frame et al., 2022b). The previous 'limitations' can thus now be seen as challenges for improving the skill of

hybrid models, such as data assimilation, model optimisation, ensembling, and hybridization, where models are merged with other ML methods (including simulations and physical models, e.g. Mosavi et al., 2018).

### 4.1 Obtaining physically realistic results

One important challenge of hybrid models is the ability of such models to produce physically-plausible or explainable results in unseen extreme conditions such as severe floods, droughts, intense heatwaves and tropical storms. Compared with physics-

based models, data-driven models were once thought to be unable to accurately predict values outside the range of the training data. However, recent work has shown this assumption is misleading. For instance, it has been shown that ML models can predict streamflow extremes that are larger than those seen in the calibration set, and better than in physics-based models (Frame et al., 2022a). As new weather records are set in different parts of the world, the challenge is ensuring that models can produce credible results under extreme forcing conditions.

One emerging route for hybrid models is to employ physics-guided or theory-guided ML designs that explicitly observe the law of conservation of mass. Such approaches seek to integrate physical knowledge within the statistical or machine learning models to take advantage of the strengths of both data-driven and physical models. For instance, Hoedt et al. (2021) created an LSTM architecture that obeys conservation laws, and these laws can also be used to guide physical interpretation of model outcomes. Although there have been considerable methodological advances in interpreting neural networks (Wilby et al., 2003;

Toms et al., 2020), physics-guided ML approaches (also referred to as physics-informed, physics-aware, or theory-guided approaches) still require further development.



Another new development is differentiable, learnable physics-based models that can approach the performance of ML models but also output internal physical variables such as evapotranspiration and soil moisture (Feng et al., 2022b). Tsai et al. (2021) first demonstrated the ability of connected neural networks to provide physical parameter sets to process-based models. They
showed the efficiency and generalizability of this paradigm for untrained variables, spatial extrapolation and interpretability. In data-sparse regions, this approach can even produce better daily metrics and future trends than LSTM (Feng et al., 2022a). These models seek to combine the power of ML and physics and have the potential to alleviate data demand, extrapolate better in space and for more extreme conditions, and be constrained by multivariate observations to enable better forecasts. Furthermore, they provide a systematic pathway for asking scientific questions and getting answers from big data.

Explainability is sometimes useful to help develop trust in model predictions. One way to achieve explainability is by providing storylines around the hybrid forecasts, to demonstrate the geophysical credibility of the results. Fleming et al. (2021) show how hydroclimatic storylines can be produced for clients to make the forecast interpretable in terms of understandable geophysical processes. They use pragmatic methods such as 'popular votes' for the candidate predictors cast by a genetic algorithm. The methods reveal how the values of predictors such as antecedent flow and snow water equivalent can help
explain the ensemble mean predicted volume.

LSTMs can also be used to post-process outputs from physics-based models, such as long-term streamflow projections (Liu et al., 2021) and streamflow simulations (Frame et al., 2021) to make them more realistic. Liu et al. (2021) implemented a physics-informed approach to post-process the streamflow projections from GCMs, GHMs and the Catchment-based Macroscale Floodplain model (CaMa-Flood). The LSTMs were trained to learn a relationship between simulated streamflow (from the
physics-based model GHMs-CaMa-Flood), basin averaged daily precipitation, temperature, windspeed and observed streamflow. The LSTM model can thus be perceived as a post-processor which aims to constrain (i.e. reduce the uncertainty of) the streamflow simulations from the physics-based model. This post-processing approach improved the simulations for the reference period, and was then successfully applied to project streamflow over the future period. However, the authors concede that this LSTM-based post-processor is still subject to the same limitations as other post-processing methods, such as the
assumption of stationarity in the parameters of the post-processing method.

Frame et al. (2021) similarly employed an LSTM to post-process the outputs from the physics-based US National Water Model (NWM). They implemented two variants of the post-processing method, alongside an LSTM forced with atmospheric inputs only (i.e. without any NWM inputs). The authors showed that the routing scheme and the land surface component of the NWM introduced timing and mass balance errors in the simulations. Thus, in some cases, it would be preferable to simply use
an LSTM model that can simulate streamflow from atmospheric forcings only (without any NWM inputs), to avoid propagating errors from the NWM to the post-processed streamflow. More broadly, the presence of data errors in observed hydroclimate records means that an unconstrained ML performs better than a physics-guided ML model because of the ability to learn and account for data errors (Beven, 2020; Frame et al., 2022b), including heteroscedastic and nonstationary data errors (Kratzert et al., 2021).



## 4.2 Assimilating human influences

One emerging challenge is assimilating human influences on the water cycle to obtain better predictions of hydroclimate variables, and especially droughts (Brunner et al., 2021; Van Loon et al., 2022). Limited data exist on human impacts such as water storage, groundwater depletion, irrigation, land cover changes, and water transfers. Therefore, how can human decisions, such as the management of reservoir levels or flow abstraction, be integrated within hydrological forecasts? This question is especially relevant over longer timescales, as well as for hydrological forecasting in general, as access to such data is limited (e.g. only very limited information on reservoir operations is included in GloFAS). One option is to develop proxies to detect and model human influence. For instance, Han et al. (2022) employed census information on the number of households to extend UK urbanisation records. Similarly, Slater and Villarini (2018) used population density data as a proxy for urbanisation and considered the extent to which seasonal streamflow predictability might benefit from 'anthropogenic' predictors such as land cover change alongside seasonal climate forecasts. López and Francés (2013) supplied a dynamic reservoir index alongside climate indices to predict historical annual maximum peak discharge in Spanish rivers. Bogner et al. (2019) introduce information on the day of the week and on local festivities as a proxy for difference in energy demand. Such proxies might also inform a hybrid system on hydro-peaking in rivers downstream from dams.

The lack of future predictions of human activities at the catchment scale is also a major limitation for hydrological forecasting over longer timescales. Here, the increasing coverage and resolution of satellite data may help to provide relevant inputs to hybrid forecasting models such as future predictions of land use change (e.g. Moulds et al., 2015). Emerging satellite altimetry products (e.g. SWOT) may enable a better understanding of reservoir operations, which can be used to constrain hydrological forecasts. Similarly, ML could potentially be used to translate major socio-economic drivers into land cover change. Overall, we suggest that the main bottleneck to integrating human activities in hybrid forecasting systems is not the model algorithms themselves, which can be adapted to any potential predictors, but rather the lack of consistent historical and future time series data on these activities.

## 4.3 Developing predictive skill

Dynamical forecasts and predictions tend to have low skill over long lead times. The skill of short-term hydroclimatological forecasts is typically constrained by the skill of meteorological forecasts, which is currently in the range of 3 to 10 days ahead but has been advancing by about one day per decade (such that "today's 6-day forecast is as accurate as the 5-day forecast ten years ago" Bauer et al., 2015, p.47). For low flows skill may currently extend up to 20 days, but this is mostly due to the quality of the information on initial conditions and the memory effect of catchment storage (Fundel et al., 2013). Seasonal climate forecasts also have low predictive skill beyond a couple of months, while decadal predictions suffer from the underestimation of atmospheric circulation in climate models, a phenomenon known as the 'signal-to-noise paradox' (e.g. Smith et al., 2020).

One of the advantages of hybrid predictions is that the statistical methods can be used to enhance predictive skill through pre- and post-processing (e.g. of the meteorological or climate forecasts). For instance, decadal predictions are skillful over



multiyear forecast periods but have too much uncertainty to provide useful information on interannual variability. Although the CMIP5-6 models can skillfully reproduce certain large-scale circulation patterns, the magnitude of teleconnections tends
to be underestimated. Statistical approaches such as 'NAO-matching' attempt to resolve this by selecting members based on their ability to reproduce climate indices and their teleconnections (Smith et al., 2020). Such methods have been employed to enhance decadal streamflow prediction (Moulds et al., 2022) and condition seasonal hydrological forecasts (Donegan et al., 2021). However, further work is still needed to interpret multiyear forecasts to provide actionable information. Given a skillful multiyear forecast, it should be possible to estimate the increased flood or drought risk (for instance) in each year of the forecast
period. Statistical and ML models may aid in future developments by trying to draw out the climate models or climate model members that perform well in given months or lead times (e.g. Slater et al., 2017).

### 4.4  Seamless forecasting: merging forecasts, predictions and projections

The utility of hybrid models for 'seamless' hydroclimatic prediction systems spanning weeks to decades is an open research question (Figure 4). There is a growing need for reliable long-term projections of climate change impacts such as floods and
droughts over the coming decades (i.e. 1-40 years ahead), yet reliable information does not exist over such timescales. The lack of seamless climate information is explained by the fact that different scientific weather and climate products have been developed for different applications. Short-term predictions (less than 5 years ahead) tend to rely more on correct initial conditions while long-term predictions and projections (>10 years ahead) rely more on correct external forcings such as greenhouse gases (Boer et al., 2016).
One way to provide longer-term climate impacts information over the coming decades is to constrain uninitialized climate model projections (e.g. climate simulations for the RCP4.5 or RCP8.5 scenarios) using initialized decadal predictions (i.e. the CMIP5-6 decadal hindcasts), which tend to better reflect observed climate variability. Befort et al. (2020) developed a method that does this by selecting the climate projections that best match the mean of the decadal predictions over the next 10 years. They showed that the constrained ensemble – which consisted uniquely of uninitialized projections for the upcoming 50 years
– had higher skill than the full projection ensemble, even after the 10-year period, once decadal prediction information was no longer available. A hybrid system for enhanced prediction of hydroclimatic impacts (e.g. flooding) could integrate the outputs of such a constrained ensemble.
   Beyond the use of uninitialized projections by themselves (covering the whole 1-50 year period), temporally concatenating bias-corrected time series of decadal climate predictions and climate projections is also possible. Befort et al. (2022) assessed
different types of bias correction and found that the variance inflation (VINF) method could reduce inconsistencies between the two time series, especially for central quantiles of the climate time series (close to the multi-model ensemble median). However, the method could not eliminate all inconsistencies, notably those for extreme quantiles. A seamless hybrid method would therefore be more difficult to generate for hydroclimate extremes such as floods and droughts. However, these two papers (Befort et al., 2020, 2022) open the way for novel research on the merging of decadal predictions and uninitialized projections
as input to seamless prediction schemes for hydroclimate impacts using hybrid ML-based approaches.





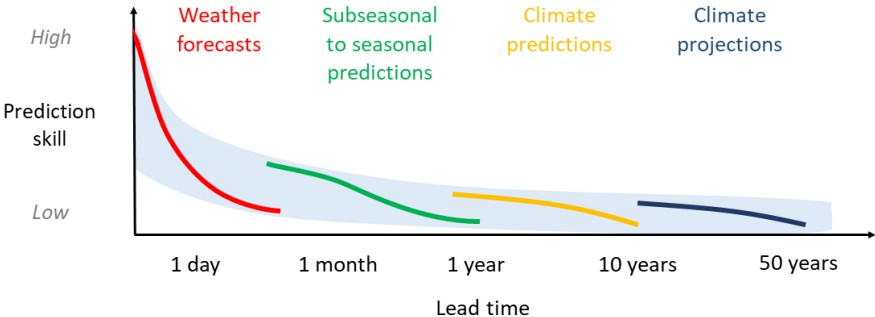

**Figure 4.** Hybrid models could be a promising route for seamlessly linking initialized predictions to scenario-based projections across timescales. Different ML-based bias-correction approaches could be explored for merging or concatenating the covariate time series (e.g. Befort et al., 2022) before using them to drive a hybrid hydroclimate prediction model (e.g. for streamflow). Such an approach is likely to be more challenging for extremes such as floods and droughts, and remains an open research question.

## 4.5  Lack of spatial variability

The data employed in many conceptual/hybrid and certain physics-based hybrid models are often lumped, i.e. spatially-averaged at the catchment scale, ignoring spatial variability in landscape and atmospheric forcing. Lumped models are challenging for the prediction of hydroclimate in complex environments such as snow-dominated watersheds, which may have karst

conduits, or spatiotemporal variation in snow accumulation and snowmelt processes. However, new approaches exist to overcome this limitation in statistical/machine learning models. For instance, Shi et al. (2015) developed a convolutional LSTM, termed convLSTM, which is able to capture spatiotemporal correlations, considering both the input and the prediction target as spatiotemporal sequences. One example is the use of past and future radar maps as input and output: such spatiotemporal sequences have high dimensionality and until recently could not be included in hydroclimate prediction schemes. Similarly,

Gupta et al. (2021) develop a spatial variability aware neural network, termed SVANN-E, in which the architecture of the neural network varies spatially across geographic locations. They evaluate the approach using high resolution imagery for wetland mapping. Such novel spatiotemporal prediction approaches are just starting to be used for hydroclimate prediction. Xu et al. (2022) used a hybrid approach to predict streamflow in a watershed with spatially variable karst carbonate bedrock. They combine a spatially-distributed snow model with a deep learning karst model based on convLSTM, which simulates the

effect of surface and subsurface properties on the streamflow. This approach allowed the authors to better include the spatial variability in the input variables to their prediction scheme.

## 4.6  Interpretability and usability

ML approaches for hydroclimate prediction over subseasonal to decadal lead times are in the early stages of development with limited operational implementation to date. Despite a commitment to develop the use of ML within operational hydrology





(Environment Agency, 2022), close co-operation is needed between the hydrology, forecasting and ML communities to explore their potential (Mosavi et al., 2018), build trust (Haupt et al., 2022), communicate skill (Thielen-del Pozo and Bruen, 2019), and overcome barriers to operational uptake (Speight et al., 2021).

In hybrid set-ups similar to the one of Humphrey et al. (2016), which requires the development of both an ML and a conceptual model, a marginal improvement of the overall streamflow forecast quality might not be worth the extra effort. For
instance, in their specific application for three gauges in southern Australia, the authors found that even though the hybrid model was more skillful than either the conceptual or the data-driven models alone, this increase of skill was only marginal for one of the three study locations (at the outlet of the basin). They concluded that for this given station, the extra time and effort required to implement the hybrid model was not worth the small gains. Implementing a hybrid modelling framework for streamflow forecasting often requires extensive time and expertise, given that two completely different types of models (here,
conceptual and data-driven) must be developed in parallel. Overall, the operational uptake of hybrid models is expected to be quicker in cases where there is no existing forecasting capability (requiring modification) or where complex physical processes make traditional approaches challenging.

## 5   Conclusions and remaining research areas in hybrid forecasting

Hybrid forecasting is emerging as a complementary approach to traditional hydroclimatic forecasting techniques but important
questions remain regarding their place in the pantheon of methods. We lay out some of the most important research possibilities.

First are questions about the evaluation of hybrid methods. How well do dynamical-statistical methods perform when compared with more traditional, operational approaches? What benchmarks should be used? How reliable are these models, and over what lead times can they be trusted? As far as we are aware, there have been very few papers (if any) comparing the skill of hybrid models with operational systems. A systematic comparison of 13 different models in a non-operational context
(including machine-learning-based, basin-wise, subbasin-based, and gridded models), however, did reveal the superiority of the data-driven LSTM-lumped model in all experiments (e.g. Mai et al., 2022), suggesting that hybrid LSTM-based prediction systems would be a promising route for future forecasts.

Second are questions about the potential for seamless prediction. To what extent can hybrid approaches be employed to meld historical trends, near-term and decadal predictions of hydroclimate variables from atmospheric forecasts, climate model
predictions, and projections? How would such a system be used operationally? Seamless hybrid prediction may provide better insights into long-term hydroclimatic trends, but merging across time-scales can lead to inconsistencies in the time series (i.e. "jumps" or step-changes) between e.g. decadal climate predictions and the climate projections (Befort et al., 2022).

Third are questions about use of data-driven models to detect and attribute the drivers of hydrologic change (Slater et al., 2021a), and then integrate such knowledge within a predictive framework. How can ML/statistical approaches be employed
to understand the relative contributions of different predictors, including human impacts such as the effects of reservoirs on streamflow (Brunner and Naveau, 2022)? To what extent can hybrid models uncover "hidden" large-scale climatic or anthropogenic drivers of change (Renard et al., 2022; Lees et al., 2022)?





An important step forward would be the development of consistent global datasets of climate hindcasts at various time scales at the catchment level. Similar datasets developed for large sample hydrological analyses such as CAMELS (Addor et al., 2017; Coxon et al., 2020) and Caravan (Kratzert et al., 2022b) have driven rapid progress in ML methods for simulating daily streamflow using observed climate inputs. Such a dataset could drive progress towards operational hybrid systems by making it easier for model developers to train and test potential methods in a pseudo-operational context. Moreover, such a dataset could integrate consistent estimates of other potential drivers – including streamflow signatures and local characteristics related to topography, geology and land cover (as in the CAMELS datasets) – enabling forecasters to understand the contribution of different drivers to streamflow predictability across time scales.

Finally, there are questions about the acceptance of hybrid models in operational contexts, given the dominance and pre-conceived superiority of physics-based forecasting and prediction methods (Cohen et al., 2019). In what ways could hybrid approaches complement, support, or replace conventional physically-based systems? The pace of change in such settings is often constrained by practicalities, institutional resistance (Arnal et al., 2020) or the requirement of decision-relevant evidence of skill. Acceptance might be advanced by systematically comparing the outputs from hybrid models with operational models under identical forcings, to assess the physical interpretation of model results (e.g. Mai et al., 2022). To convince operational forecasters that hybrid models may provide added value alongside more traditional approaches requires rigorous benchmarking by the community alongside established approaches.

There are several possible paths forward. One of these paths frames hybrid models not as a replacement of current operational systems but as a complementary tool, helping on different levels, and likely within existing systems. Another path forward is to recognize the skill delta between ML and hybrid models vs. classical models, and to start to develop future replacements for current operational models; replacements based fundamentally on AI principles, but with the ability to incorporate elements of traditional hydrological and climate science where such are beneficial. Furthermore, hybrid models could be developed to estimate both impacts and mitigation measures, based on past events. All of these approaches make sense for different reasons and in different scenarios, and various agencies and organizations are pursuing both these and other strategies for incorporating ML into operational modeling workflows. Overall, the utility of hybrid models is not only for enhancing forecasting and prediction, but also for allowing deeper interrogation of diverse data, revealing sometimes hidden or obscure hydrological processes.

*Author contributions.* All authors contributed to writing the review.

*Code availability.* There is no code or data associated with this review article.

*Competing interests.* At least one of the (co-)authors is a member of the editorial board of Hydrology and Earth System Sciences.



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

*Acknowledgements.* LJS acknowledges support from UK Research and Innovation (MR/V022008/1) and NERC (NE/S015728/1). AYYC and MZ acknowledge support from the WSL MaLeFiX project within the program "Extremes". LA is supported by the Canada First Research Excellence Fund's Global Water Futures programme. GV acknowledges support from the USACE Institute for Water Resources. CM acknowledges support from Science Foundation Ireland (SFI/17/CDA/4783).