# Peer review of "Hybrid forecasting: blending climate predictions with AI models"

_Hydrology and Earth System Sciences, 2022_

## Author Comment (AC1)

**Hybrid forecasting: using statistics and machine learning to integrate predictions from dynamical models**

**Response to Reviewer 1**
**Anonymous Referee #1, 20 Oct 2022**

Thank you for the opportunity to review Slater et al. "Hybrid forecasting: using statistics and machine learning to integrate predictions from dynamical models". Overall, I find this to be a timely and informative review. However, I do have a variety of comments, detailed below. I recommend at least a minor revision, if not a major revision.

We are grateful to Reviewer 1 for their positive and helpful comments on our manuscript. Their comments are copy-pasted below verbatim in black font, and our replies are in blue font. We label the comments in the following manner: "R1.C1" indicates Reviewer 1, Comment 1.

**R1.C1:** My biggest concern in reading this paper is the number of different models and approaches etc. that are discussed. The paper is full of acronyms (so Table 2 is certainly helpful) such that I routinely found myself lost in the details and trying to remember the bigger picture or category that the details were supporting. If I'm someone coming to this review trying to figure out where to start with hybrid modeling, I think I would really struggle. How would I begin? Would I choose a model/paper from Table 1? How would I discriminate or know how to choose among the myriad of options? If the authors can provide some answers or guidance to these types of questions, I think it would be very helpful. Also, if there is any way to more clearly emphasize the main points even among all the details.

We appreciate this opinion, and we agree with the Reviewer that the paper should provide a clear introductory overview of the different types of hybrid models for someone new to the field. We propose to add a new paragraph that more clearly outlines where a user could begin (depending on their aim) and articulates the main characteristics of each hybrid model type. We propose to base this paragraph on an expanded and improved version of the current Table 3 (see response to **R1.C17**).

**R1.C2:** Terminology is really important in this paper. Can you please provide some definitions of the differences between physics-based vs. conceptual models?

Yes, we agree entirely. We have now included definitions of physics-based, conceptual, and dynamical models upfront in the first paragraph of the revised manuscript.

**R1.C3:** One question I had was whether any hybrid schemes are currently operational. But, this is partially answered in line 93. Also wanted to see what the authors think it would take to make these models operational, which is partially addressed in the conclusion. Any further details that can be provided on this topic would be greatly appreciate (i.e., are there ANY examples of operational hybrid schemes? And if so, can they serve as pilot projects? i.e., what can we learn from their implementation that might help hybrid schemes become more widely used?).

We appreciate this is an important point and we will add discussion of operational hybrid schemes in the revised manuscript. For instance, the US Climate Prediction Center runs an objective consensus climate forecast that uses ensemble regression to combine multiple dynamical and statistical forecasts into one. The International Research Institute for climate and society (IRI) has a multi-model calibrated prediction based on three SubX models. Lastly, the Google flood forecasting model is also operational. We will include a discussion of what can be learnt from the implementation of these hybrid schemes.

Beyond these operational hybrid examples, there are also cases of hybrid forecasting where the statistical part of the forecast is run separately by a stakeholder using dynamical forecast outputs from the producing centre -- these examples are not always visible as a single 'hybrid' activity but are operational nonetheless. We will include some examples.

Finally, one important point that we will discuss more explicitly in the revised text is that almost all climate-scale hydroclimate projection is hybrid. This is because some kind of statistical processing is almost always applied to an ensemble of CMIP outputs (though the projections may not necessarily be 'operational').

**R1.C4:** Lines 100 and on list many hybrid models… but not all the references are in Table 1 as well. Any reason? (e.g., Miller et al., 2021)

Initially, we only presented a representative selection of models in Table 1 because we were concerned that it might become a very long table. In the revised manuscript, we will add further key hybrid papers that are discussed in the manuscript and we will justify our selection.

**R1.C5:** Section 2.4 seems to have a different focus than what is indicated on line 122.

Thank you for spotting this; the text has been revised accordingly ("and hybrid forecasts including a conceptual hydrological model").

**R1.C6:** The grammar of the sentence spanning lines 122-124 isn't quite correct. Same for the sentence spanning lines 273-274.

Thank you - both sentences have been updated.

**R1.C7:** Lines 243: seems like a concluding statement (summarizing the overall point of the paragraph) is needed here.

We agree, and a concluding summary statement will be added.

**R1.C8:** Line 249: the reference to Madadgar et al., 2016 – where was this study applied?

The study was applied to the southwestern United States. The text has been updated to reflect this.

**R1.C9:** Lines 264-266: Is this sentence a description of "mode-matching"? And if so, can that be made clear. If not, please provide a brief idea of what mode-matching is.

We have included a definition and updated the citations.

**R1.C10:** Line 409: by "national" does that mean the United States?

There are different CAMELS datasets for different countries, including the United States, United Kingdom, Chile, Brazil, Australia, France, and Switzerland (available soon). We will clarify which countries these datasets are available for, in the revised text.

**R1.C11:** Line 440: what does "surface water" mean?

The term "surface water" at line 440 referred to a paper by Rözer et al. (2021) on pluvial flood forecasting. We will clarify the text.

**R1.C12:** Lines 454-461: this paragraph, especially the last sentence, seems to imply there are no limitations to hybrid models.

Thank you - we did not intend to give this impression and will revise the wording. The sentence "*The previous 'limitations' can thus now be seen as challenges*" suggested that they are no longer limitations. We will reframe this point to show that the limitations *"are indeed challenges…"*, and we will ensure the limitations are more clearly highlighted in the revised manuscript. These include, for instance, challenges related to physics-guided ML designs, difficulties in assimilating human influences, or the 'curse(s) of dimensionality' (problems of data sparsity, multicollinearity, multiple testing and overfitting) mentioned by Reviewer 2.

**R1.C13:** Lines 491-509: are these paragraphs in the correct place? The information presented within seems to go in Section 2.1 on pre- and post-processing.

These two paragraphs have been moved to section 2.1, and will also be shortened a little, as that section is now quite long.

**R1.C14:** Lines 598-599: this is a really important point that I'm glad was made (i.e., the marginal improvement might be not worth the effort). It seems to me that dealing with this issue is critical to making hybrid schemes more widely accepted. Is there any way we can determine a priori the marginal improvement (without having to build both models in parallel and then compare)? For example, the Mai et al. (2022) study in line 616 – would be good to comment if the demonstrated superiority was enough to justify the extra effort.

Yes, we agree that this is an important but tricky point, as it is hard to know *a priori* how much added value can be obtained without first building a hybrid model and benchmarking the results. We agree that the Mai et al. (2022) study is the first of its kind in providing such a detailed intercomparison of modelling approaches, and it suggests the effort related to using ML is justifiable. Although the findings do help sway the field, they are for

simulation, rather than forecasting (for which there are more steps needed). In other words, it is still a jump to speculate that ML provides improvements for prediction.

Another important point is that the development of successful hybrid methods for short-term forecasting does not automatically imply success for medium range, subseasonal-to-seasonal, or decadal forecasting. In the revised text, we will include a discussion of various issues associated with the implementation of operational hybrid schemes, such as the shift of expertise, and the potential shift of computing architecture when implementing GPU-requiring ML techniques. We will more clearly discuss the fact that a lot of hybrid operational forecasting is currently being implemented, as described in the response to **R1.C3**. In many cases, the dynamical producing centre draws a line before the 'tailoring' statistical part, which the stakeholders implement.

**R1.C15:** Table 1: (a) Are any of these operational? (b) Any rationale for inclusion/exclusion of studies in this table? (c) Can you add another column that describes how the statistical and dynamial models are combined? (d) Regarding column headings, in the text, "data-driven" seems to be the most generic term (lines 25-26) but here the column header is "statistical" model (and elsewhere, "empirical" is used). Again, the importance of terminology in this paper. (e) Would this table become slightly easier to digest if it was first sorted by predictand type (i.e., streamflow vs. reservoir, etc) and then horizon? I'm not sure, but I think that predictand is a larger category (and what I would first be interested in), then horizon.

(a) We will include a number of operational examples, such as those mentioned in our reply to **R1.C3**. We are revising the text to reflect that hybrid hydroclimate forecasting is a form of operational practice already.

(b) We sought to cover different types of dynamical and statistical models, different ranges, and different variables. Including all the available papers would be too many, but we will ensure that we have a representative sample of all the different study types (and will attempt to provide a classification of the types that exist).

(c) This is an excellent suggestion. Depending on space, we will either add a column in Table 2, or provide further examples in Table 3 (which we plan to develop; see response to **R1.C17**).

(d) The column heading has been updated to "data-driven".

(e) We have made the choice to sort by horizon, but will assess how the table looks when sorted by predictand then horizon, and make a decision. Thank you for the useful suggestions!

**R1.C16:** Some acronyms that are not defined anywhere: RCP8.5, FV3GFS (this is just the name of the atmospheric model?), PREVAH (also a model name?)

The definitions of these acronyms have all been added to Table 2: Representative Concentration Pathway 8.5 (high-emissions warming scenario); Finite-Volume Cubed-Sphere Global Forecast System (global atmospheric model); and Precipitation-Runoff-Evapo-transpiration Hydrotope Model. We will also check the text to make sure we have not accidentally missed any other acronyms.

**R1.C17:** Table 3: (a) Shouldn't "coupled" be included here also, since it is discussed in the text. (b) I find it interesting that Lee et al. (2002) is a primary reference for two of the options (serial and parallel) – given that it is now 20 years ago. Is that because it was such a foundational paper? Either way, can a more recent reference also be provided? As a corollary comment: It would be nice to have a discussion in the text of when these approaches were first tried (what was the foundational paper) on hydroclimate variables.

We propose to update Table 3 to provide a more comprehensive overview of the different types of hybrid structure that exist and will use this revised table to better clarify and describe the approaches in the main text.

(a) We will update Table 3 to include the 'coupled' approach, and we will make sure the terms 'coupled' and 'parallel' are more clearly defined (for instance, we will distinguish the term 'coupled' from coupled dynamical models and clarify that one-way post-processing from dynamical to statistical is not coupling).

(b) Lee et al. (2002) happened to use and compare both those terms, but we agree that more recent references should be added in here, and will update the table accordingly.

We will also add and discuss foundational papers, such as Glahn and Lowry (1972) on postprocessing using model output statistics (MOS). One challenge is that different terminology is used in different fields, and the approach is not always referred to as "hybrid forecasting", so it can be difficult to find older papers.

Glahn, H. R., & Lowry, D. A. (1972). The use of model output statistics (MOS) in objective weather forecasting. Journal of Applied Meteorology and Climatology, 11(8), 1203-1211.

**R1.C18:** Figure 1: A few comments/questions on this graphic: (a) Please explain if the coloration of the boxes has any meaning. (b) Aren't large-scale predictors etc. also inputs to the hybrid forecasting scheme (not just dynamical predictors) – in other words, the straightforward left-to-right is not actually quite so straightforward? (c) Bottom middle: shouldn't it be "hydroclimate model" rather than "hydrological model" to be more general?

Thank you for helping us make the figure more intuitive.

(a) The colour of the boxes indicates the broad type of prediction scheme and serves to help the reader see how the top two schemes (rows) are combined in the third scheme (bottom row, reflecting hybrid prediction); we have clarified the figure caption accordingly (please see revised caption below).

(b) Yes, large-scale predictors can also be used as inputs, but would likely be issued from dynamical predictions or dynamical reanalyses (e.g., using large scale principal components to identify predictors) in the case of a hybrid forecast (although some observations might be employed too). We will update the figure caption to make it clear that multiple different types of combinations are possible.

(c) Yes, we agree that the bottom middle box would be better with the term "hydroclimate model" and have updated it accordingly; thank you for spotting this.

[Figure]

Figure 1. Defining hybrid hydroclimate forecasting and prediction. "Hydroclimate" refers to a range of variables defined in the text (including streamflow). The top row indicates traditional dynamical hydroclimate predictions (blue); middle row is data-driven predictions (yellow) and bottom row is hybrid predictions (red, with three examples of hybrid structure from top to bottom: statistical-dynamical, serial, and parallel, with an example of ML-based post-processing of dynamical model output in the bottom left box). The figure provides a simple example, but more complex schemes are possible, including e.g., a mix of observations and predictions in the left column.

**R1.C19:** Figure 2: So, you obtain one value each for JJA, then take the max? Could be clarified in the caption text. The maximum summer discharge is the largest of the 92 daily values in the June-July-August period. The caption has been revised to state this explicitly.

Thank you for this constructive review!

---

## Author Comment (AC2)

**Hybrid forecasting: using statistics and machine learning to integrate predictions from dynamical models**

**Response to Reviewer 2**
Anonymous Referee #2, 09 Nov 2022

**Summary**

This paper reviews - indeed it defines - the burgeoning field of hybrid dynamical-statistical hydrometeorological forecasting. The paper is timely and I believe it to be of wide interest to readers of HESS (and very likely beyond). I generally like to balance positive and negative feedback in reviews, but it was very difficult for me to find any suggestions to improve in this paper. It is skillfully organised, placing a very wide range of studies in sensible categories and highlighting specific themes with more detailed discussions of some papers. I didn't think there were really any major gaps in the literature and ideas they presented. The paper is also brilliantly written, with concise, lucid sentences making it an easy read - I believe even for non-experts. In short, in my view this review does everything a review should do: summarises the literature comprehensively, shapes the literature sensible themes, makes an argument - in this case the paper is essentially arguing for the recognition of hybrid forecasting as a distinct field (or at least a subfield within hydrometeorological forecasting) - and makes clear recommendations on the future direction of hybrid forecasting. I congratulate the authors on a remarkable review paper, one that I believe deserves to be widely cited.

Reply. We are most grateful to the Reviewer for this kind assessment of our work! The Reviewer's comments are copy-pasted below verbatim in black font, and our replies are in blue font. We label the comments in the following manner: "R2.C1" indicates Reviewer 2, Comment 1.

**Specific comments**

**R2.C1:** L33 "We do not provide a prescriptive definition of hybrid forecasting as it exists along a continuum, which may include a wide range of modeling and 'big data' type Earth Observation (EO) datasets" Fair enough - a sensible choice.

We are glad the Reviewer agrees with this choice!

**R2.C2:** L156 "ML models are also employed during the dynamical climate model simulations to correct model biases" I suspect the use of 'ML' to describe Bayesian techniques like Schepen and bias-correction methods like Meyer may be a bit unusual to many. Suggest the broader term 'statistical models' or 'data driven models' (consistent with the definition given in the introduction) to encompass all these.

We have updated this to "data-driven models".

**R2.C3:** L156 "The use of ML..." same issue with this paragraph - I would say that neither Bennett et al. nor McInerney et al. really qualify as ML - they are error models, which I think in general usage don't get lumped in with ML. These distinctions may well be arbitrary, but I'd suggest if the authors want to broaden the common use of ML to include a wide range statistical models that this be defined up front somewhere (in the way the authors have done with 'data-driven').

Fair enough - we have updated this paragraph to "data-driven models" also.

**R2.C4:** L453 "4 Key challenges and opportunities of hybrid forecasting" I guess I would add to the topics covered in this section the effective use of probabilistic forecasts in decision making. One of the major efforts in hybrid forecasting systems has been to achieve reliable predictive distributions; but it's not yet clear that this effort will necessarily result in better decisions. It's likely that automated decision systems/optimisation will be the means to take advantage of reliability in ensemble distributions. In my view this still requires considerable research - existing methods of optimsation do not necessary take advantage of this property. But I also understand that this may be outside the scope of what the authors wish to address - the paper is really comprehensive in the areas it does choose to address, so they may feel they cannot do this area justice (even if they agree that it is worth discussing). I will leave it to the authors to decide whether this is worth including in their paper.

We entirely agree with the Reviewer that the development of probabilistic forecasts and their subsequent uptake in decision making (and potential for improving decisions) is an important topic to address. In revising our manuscript, we will review the existing literature and see if it is feasible to include this point in the section on Challenges and Opportunities.

**R2.C5:** L456 "ML models include the requirement for large datasets (previously discussed)" This review presents the availability of large datasets for ML as a strength of ML - which it of course is - but it presents few of the difficulties associated with using these datasets for prediction, for example some of the 'curse(s) of dimensionality' described by Altman & Krzywinski (2018). ML models are still subject to some of these issues - though I realise canvassing these is not the main aim of the paper. Whether these matters are best discussed in this paper is a subjective judgment: I am happy to defer to the authors on this point.
This is a nice suggestion. We will attempt to include a couple of sentences on the difficulties associated with the use of large datasets for hybrid prediction, based on this reference. There may also be difficulties applying some techniques for subseasonal-to-seasonal (S2S) prediction that are not an issue for shorter range (and longer) forecasting because the S2S sample sizes can be so much smaller (often nearer 100 versus thousands for shorter ranges).

**R2.C6:** L465 "data-driven models were once thought to be unable to accurately predict values outside the range of the training" I'm not sure this is really true (or if it is, I haven't been exposed to it) - would be good to provide a reference in support of this statement. There is a long history of statistical extrapolation - not least in extreme value theory or design engineering - for exactly these purposes.
It is interesting that there seem to be different opinions on the question of data extrapolation by data-driven models. We will carefully review the literature on this point and revise the sentence accordingly (either delete it or provide references).

**R2.C7:** L487 "Explainability is sometimes useful to help develop trust in model predictions" this is a very interesting point - in my experience forecasting agencies frequently engage in this kind of story-telling, both for internal and external communications, so this is probably an important box to tick for the widespread adoption of hybrid forecasting systems. I'm not suggesting any change here, but I guess I also feel this kind of narrative building can be antithetical to the effective use of (usually carefully constructed) probability distributions that come out of hybrid forecasting systems.
This is an interesting point for discussion too. We will attempt to update the text to reflect the different perspectives on the use of storytelling/narratives versus the use of probabilistic forecasts, based on the literature. One reason for providing explainability of a prediction model is that when predictions evolve from forecast to forecast for a given variable (e.g. spring runoff), stakeholders want to know why (i.e. what has changed). This explainability can be important, but there are indeed cons, and we will discuss these in the revised manuscript.

**R2.C8:** L536 "For low flows skill may currently extend up to 20 days, but this is mostly due to the quality of the information on initial conditions and the memory effect of catchment storage" this statement may be true specifically for the study by Fundel et al. 2013, but it is phrased more generally. It is quite possible to get forecast skill of streamflow well beyond twenty days - even with simple ESP methods - (depending on catchment, time of year, etc.) so I think the authors should avoid a statement that posits a general limit on the prediction of streamflow of 20 days. Please reword this so that it is clear that this finding was specific to Fundel et al.
We have reworded this sentence so it is clear the finding is specific to Fundel et al., and that skill can be obtained beyond 20 days in other cases. Thank you.

**R2.C9:** Fig 4: As you've used 'prediction' generically in the vertical axis label ('Prediction skill') - implying (correctly in my view) that all the models in this plot produce predictions - I suggest changing the label "Subseasonal to seasonal predictions" to the more specific "Subseasonal to seasonal forecasts" and the label "Climate predictions" to "Multi-year climate forecasts".
These are excellent points, thank you very much! We have revised the figure accordingly, as shown below.

[Figure]

**Typos etc.**

L50 "While conceptual hydrological models..." suggest a paragraph break before 'While'
Done.

L71 Suggest paragraph break before 'Historically...'
Done.

L83 "to understand to which" typo - delete second 'to'
The sentence has been rewritten for clarity.

**References**

Altman N, Krzywinski M. 2018. The curse(s) of dimensionality. Nature Methods 15: 399-400. DOI: 10.1038/s41592-018-0019-x.
Thank you for the reference, which has been added to the revised manuscript.

Thank you for the helpful review!

---

## Author Response (AR1)

**Response to Editor and Reviewers**
**Hybrid forecasting: combining dynamical predictions with data-driven models**
**23 Feb 2023**

**Editor decision: Publish subject to revisions (further review by editor and referees)**
**by Nadav Peleg, 09 Jan 2023**

Comments to the author:
Dear Louise Slater,
Thank you for uploading your replies to the comments and suggestions made by the referees. Both reviews are rather positive and I agree with their assessment that the manuscript will be of interest to the readers of HESS. I invite you to upload a revised version of the manuscript, along with a point-by-point reply to the comments of the reviewers. I am looking forward to receiving the revised text.
Sincerely,
Nadav Peleg

Dear Nadav Peleg,
We are grateful for the opportunity to revise our manuscript. We hereby submit our revised version of the manuscript, along with a point-by-point reply to the Reviewers' comments (below). The Reviewers' comments are copy-pasted verbatim in black font, and our replies are in blue font. Revised text is indicated in red font. Each comment is numbered, so for example "R1.C1" indicates Reviewer 1, Comment 1. We have also edited the manuscript title slightly for clarity.
Sincerely,
Louise Slater, on behalf of the co-authors.

**Response to Reviewer 1**
**Anonymous Referee #1, 20 Oct 2022**

Thank you for the opportunity to review Slater et al. "Hybrid forecasting: using statistics and machine learning to integrate predictions from dynamical models". Overall, I find this to be a timely and informative review. However, I do have a variety of comments, detailed below. I recommend at least a minor revision, if not a major revision.
We are grateful to Reviewer 1 for their positive and helpful comments on our manuscript. Their comments are copy-pasted below verbatim in black font, and our replies are in blue font. We label the comments in the following manner: "R1.C1" indicates Reviewer 1, Comment 1.

**R1.C1:** My biggest concern in reading this paper is the number of different models and approaches etc. that are discussed. The paper is full of acronyms (so Table 2 is certainly helpful) such that I routinely found myself lost in the details and trying to remember the bigger picture or category that the details were supporting. If I'm someone coming to this review trying to figure out where to start with hybrid modeling, I think I would really struggle. How would I begin? Would I choose a model/paper from Table 1? How would I discriminate or know how to choose among the myriad of options? If the authors can provide some answers or guidance to these types of questions, I think it would be very helpful. Also, if there is any way to more clearly emphasize the main points even among all the details.
We greatly appreciate this opinion, and we agree with the Reviewer that the paper should provide a clear introductory overview of the different types of hybrid models for someone new to the field. We have therefore fully re-written the opening paragraphs of the paper to outline three main hybrid model types and the characteristics of each type, alongside an expanded and clarified Table 1 (former Table 3). We have also clarified the text at various points throughout the manuscript, and have fully restructured Section 2 to better illustrate the three main hybrid model types.

"While challenging to identify distinct categories, given the flexibility and diversity of hybrid methods, three principal types of hybrid model structure may be discerned (Figure 1; Table 1). (i) Statistical-dynamical models typically drive a statistical or ML model (data-driven) with dynamical weather or climate model outputs from numerical weather prediction (NWP) models or Earth System Models (ESMs). The statistical-dynamical structure is the most common type of hybrid model in the literature (Table 2). (ii) Serial models combine data-driven and dynamical models sequentially, and may include additional types of models such as a hydrological model. (iii) Coupled or parallel approaches combine data-driven and dynamical models in parallel. The coupled approach is more commonly employed in operational settings, where ML is increasingly being used to upgrade components within existing modelling schemes.."

Table 1. Examples of different hybrid model structures.

| Name | Description |
| --- | --- |
| Statistical-dynamical | Statistical-dynamical hybrid models consist of driving or conditioning a data-driven model with dynamical weather, climate, or Earth System Model (ESM) predictions (e.g. Vecchi et al., 2011; Slater and Villarini, 2018). Both expressions 'statistical-dynamical' and 'dynamical-statistical' are used depending on the focus of the research or the field of study. This approach is also referred to as 'parameter informed' (e.g. Schlef et al., 2021) or 'physical–statistical' (e.g. AghaKouchak et al., 2022) prediction. |
| Serial | A serial structure combines the dynamical and data-driven models sequentially, and may include additional models such as a hydrological model. For instance, one could pre-/post-process the output of a dynamical model using a data-driven approach (e.g. Glahn and Lowry, 1972) and use those predictions as input to a conceptual or physics-based model. In Bennett et al. (2016), post-processed General Circulation Model (GCM) forecasts are used to force a monthly rainfall-runoff model. In Richardson et al. (2020), weather patterns are identified in an ensemble prediction system and subsequently used to forecast threshold exceedance probabilities of extreme precipitation and flooding. |
| Coupled or Parallel | In a coupled hybrid structure, the data-driven and dynamical model are combined in parallel. This may involve, for instance, replacing a component of a dynamical model with a data-driven model, e.g. to create a machine-learning corrected GCM (e.g. Watt-Meyer et al., 2021). Alternatively, it is possible to combine outputs from an ensemble of dynamical and statistical predictions run in parallel (e.g. Madadgar et al., 2016). A data-driven model may also be employed to combine dynamical predictions from both meteorological and hydrological models (e.g. Bogner et al., 2019) |

**R1.C2:** Terminology is really important in this paper. Can you please provide some definitions of the differences between physics-based vs. conceptual models?

Yes, we agree. We have now included definitions of physics-based and conceptual models in the opening paragraphs of the revised manuscript:

"Traditional workflows in which a physics-based or conceptual land/hydrology model generates the final forecast product are still the most commonly used operational forecasting systems worldwide. These may include 'physics-based' models, based on a spatially-distributed representation of known physical laws through mathematical equations and numerical solution (e.g. Freeze and Harlan, 1969), or 'conceptual' models, which simplify the representation of physical processes, often using empirical relationships (e.g. Nash and Sutcliffe, 1970)."

**R1.C3:** One question I had was whether any hybrid schemes are currently operational. But, this is partially answered in line 93. Also wanted to see what the authors think it would take to make these models operational, which is partially addressed in the conclusion. Any further details that can be provided on this topic would be greatly appreciate (i.e., are there ANY examples of operational hybrid schemes? And if so, can they serve as pilot projects? i.e., what can we learn from their implementation that might help hybrid schemes become more widely used?).

We appreciate this is an important point and we have added discussion of operational hybrid schemes in the revised manuscript as follows.

"Some notable examples of operational hybrid prediction include the 'objective consensus' climate forecast (i.e. derived objectively from multiple models) at the US Climate Prediction Center, which uses ensemble regression (e.g. Unger et al., 2009) to combine multiple dynamical and statistical forecasts into one. The International Research Institute for Climate and Society (IRI) has a multi-model calibrated prediction based on three Subseasonal Experiment (SubX) models (Pegion et al., 2019). The UK Met Office uses a tool called "Decider" which assigns medium-range precipitation forecast ensemble members to a set of 30 probabilistic weather patterns (Neal et al., 2016) and then feeds several downstream forecasting applications, such as for coastal

flooding (Neal et al., 2018) and fluvial flooding (Richardson et al., 2020). Lastly, the Google flood forecasting model (https://sites.research.google/floods/) produces operational, public-facing forecasts of water levels up to six days ahead (Nevo et al., 2022) using ML models forced with operational, real-time weather forecasts from the ECMWF Atmospheric Model high resolution 10-day forecast (ECMWF HRES) as inputs. Broadly speaking, many hydroclimate projection systems are now hybrid, as per the 'serial' definition in Table 1, because some kind of statistical processing is applied to generate a final information product from an ensemble of climate model outputs. Dynamical modelling centres often lack the resources or scope to tailor outputs to particular stakeholder needs (adding value with data-driven methods), leading to implementation of such processing by the end users themselves. These predictions are not always visible as 'hybrid' activity but are operational nonetheless. These examples show the general evolution of the field from traditional forecasting (Cohen et al., 2019) toward hybrid prediction."

**R1.C4:** Lines 100 and on list many hybrid models… but not all the references are in Table 1 as well. Any reason? (e.g., Miller et al., 2021)

We aimed to present a representative selection of models covering different predictands (river stage, precipitation, streamflow, drought, storms, reservoir inflow, surface water levels), data-driven models, dynamical models, types of hybrid model, and forecast horizons. We have revised the table carefully and added some additional examples representing the "coupled" hybrid model type especially. We also removed the paper by Miller et al. (2021) because it presents a statistical forecast rather than a hybrid forecast.

**Table 2.** Examples of hybrid forecasts of different hydroclimate variables and model types. Each example includes both a data-driven model and a dynamical weather or climate model. Examples are sorted by increasing time horizon. Hybrid model types are defined in Table 1 and acronyms are defined in Table 3.

| Predictand | Data-driven model | Dynamical model | Hybrid type | Horizon | Citation |
|---|---|---|---|---|---|
| River stage and inundation | LSTM | ECMWF HRES | Stat-dyn | 1-6 days | Nevo et al. (2022) |
| Daily streamflow | BNN, SVR, GP, MLR | NOAA GFS | Stat-dyn | 1-7 days | Rasouli et al. (2012) |
| Precipitation | RF | FV3GFS | Coupled | 1-10 days | Watt-Meyer et al. (2021) |
| Precipitation extremes and flooding | Probability of exceeding thresholds | UKMO GloSea5; ECMWF | Serial | 15 days | Richardson et al. (2020) |
| Biweekly temperature and precipitation | PLSR | CFSv2 | Serial | 2–3 & 3–4 weeks | Baker et al. (2020) |
| Seasonal streamflow | PCR & CCA | CFSv2 & ECHAM4.5 | Stat-dyn | 1 month | Sahu et al. (2017) |
| Monthly reservoir inflow | RF, GBM, ELM, M5-cubist, elastic net | FLOR | Stat-dyn | 1 month | Tian et al. (2021) |
| Drought: seasonal SPI | Dynamic-LSTM | ECMWF SEAS5 | Stat-dyn | 3 months | Wu et al. (2022) |
| Seasonal tropical storm frequency | MLR | UKMO Glosea5 | Stat-dyn | 3 months | Kang and Elsner (2020) |
| Seasonal rainfall | ANN, MLR | UKMO GloSea5, ECMWF SEAS5 | Stat-dyn | 1-4 months | Golian et al. (2022) |
| Drought | Bayesian model based on copula functions | NMME (8 models) | Coupled | 3-5 months | Madadgar et al. (2016) |
| Accumulated seasonal reservoir inflow | SVR, GP, LSTM, NLANN, DL | CMCC | Serial + stat-dyn | 1-6 months | Essenfelder et al. (2020) |
| Discharge and surface water levels | MLR, LR, DT, RF, LSTM | ECMWF SEAS5; EFAS hydrological forecasts | Stat-dyn | 1-7 months | Hauswirth et al. (2022) |
| Hurricane frequency and intensity | GAMLSS | NMME (6 models) | Stat-dyn | 1-9 months | Villarini et al. (2019) |
| Seasonal runoff | PCR | NMME (7 models); ECWMF SEAS4 | Stat-dyn | 4-9 months | Lehner et al. (2017) |
| Hurricane frequency | Statistical emulator of dynamical atmospheric model | GFDL–CM2.1; NCEP–CFS | Stat-dyn | 1-10 months | Vecchi et al. (2011) |
| Seasonal streamflow | GAMLSS | NMME (8 models) | Stat-dyn | 1-10 months | Slater and Villarini (2018) |
| Monthly streamflow | FoGSS, CBaM | POAMA-M2.4 | Serial | 1-11 months | Bennett et al. (2016) |
| Seasonal flood magnitude | GAMLSS | 5/8 CMIP 5/6 GCMs | Stat-dyn. | 2-5 years | Moulds et al. (2021) |
| Seasonal flood counts | Poisson regression | 9/14 CMIP5 GCMs | Stat-dyn | 1-10 years | Neri et al. (2019) |
| Daily streamflow | TCNN (& others) | 4 GCMs from LOCA (CMIP5) | Serial + stat-dyn | Decades | Duan et al. (2020) |
| Flood magnitude | LSTM (+5 GHMs) | 5 GCMs from ISIMIP-FT (CMIP5-6) | Serial | Decades | Liu et al. (2021) |
| Daily streamflow | DNN-PCE | 10 GCMs (CMIP5) | Serial | Decades | Zhang et al. (2022) |

**R1.C5:** Section 2.4 seems to have a different focus than what is indicated on line 122.
Thank you for spotting this; the opening sentence of Section 2 has been revised.

**R1.C6:** The grammar of the sentence spanning lines 122-124 isn't quite correct. Same for the sentence spanning lines 273-274.
Thank you - both sentences have been updated.
1) "The atmospheric and climate model predictions employed within hybrid models can range from single climate models to large multi-model ensembles"
2) "Humphrey et al. (2016) used a combination of historical observations and downscaled dynamical forecasts of rainfall and PET in southern Australia…"

**R1.C7:** Lines 243: seems like a concluding statement (summarizing the overall point of the paragraph) is needed here.
We have added a concluding summary statement:
"This example shows how a simple statistical model can be used to produce sub-seasonal to seasonal streamflow forecasts. The skill of such a scheme might be improved by post-processing the ensemble of climate predictions used to drive the model."

**R1.C8:** Line 249: the reference to Madadgar et al., 2016 – where was this study applied?
The study was applied to the southwestern United States. The text has been updated to reflect this.
"used to successfully predict seasonal precipitation anomalies in the southwestern USA"

**R1.C9:** Lines 264-266: Is this sentence a description of "mode-matching"? And if so, can that be made clear. If not, please provide a brief idea of what mode-matching is.
We have included a definition and updated the citations.
"Decadal forecast skill can be increased by 'mode-matching', which consists of sub-selecting the individual members from a large climate model ensemble of decadal predictions that best represent the multiyear temporal variability in a relevant large-scale mode of climate variability (Smith et al., 2020; Moulds et al., 2022)."

**R1.C10:** Line 409: by "national" does that mean the United States?
There are different CAMELS datasets for different countries, including the United States, United Kingdom, Chile, Brazil, Australia, France, and Switzerland (available soon). The sentence has been updated to "Such models may start to emerge with the growing availability of observational training datasets, such as the national `CAMELS' datasets (available for the United States, United Kingdom, Chile, Brazil, Australia, France, and soon Switzerland, e.g. Newman et al. 2015, Addor et al. 2017, Coxon et al. 2020) and international `Caravan' streamflow dataset (Kratzert et al. 2022)." – but we have not included all the citations because this would mean 8 citations for one sentence (and the paper is already very long).

**R1.C11:** Line 440: what does "surface water" mean?
The term "surface water" at line 440 referred to a paper by Rözer et al. (2021) on pluvial flood forecasting. The text has been clarified to indicate that this manuscript refers to pluvial floods.

**R1.C12:** Lines 454-461: this paragraph, especially the last sentence, seems to imply there are no limitations to hybrid models.
Thank you - we did not intend to give this impression and have revised the paragraph as follows:
Frequently-cited limitations of ML models include the requirement for large datasets and issues associated with the `curse of dimensionality', namely data sparsity (i.e. when there are too few data points relative to the number of dimensions), multicollinearity of the variables, multiple testing (leading to an increased number of false positives), and overfitting (Altman and Krzywinski, 2018). There is also the difficulty of obtaining physically plausible results for previously `unseen' extremes that are larger than those seen in the observational record; however, new research suggests that ML models may provide results that are more physically plausible than physics-based and conceptual models when data are biased (Frame et al., 2022b). Further challenges for improving the skill of hybrid models include data assimilation, physics-guided ML designs, assimilation of human influences, model optimisation, ensembling, and hybridization, where models are merged with other methods (including simulations and physical models, e.g. Mosavi et al., 2018). While some of the difficulties associated with large sample sizes apply less for seasonal to decadal hybrid forecasting, where the sample sizes can be much smaller (often near 100 values) than the sample sizes for shorter ranges (thousands or more), the small

sample sizes present a challenge for model training. Thus, a range of different challenges may apply depending on the forecasting horizon and data required."

**R1.C13:** Lines 491-509: are these paragraphs in the correct place? The information presented within seems to go in Section 2.1 on pre- and post-processing.
These two paragraphs have been moved to the new section 2.2.1 on "Serial pre- and post-processing of hydroclimate predictions using data-driven approaches" (Section 2 has been restructured for greater clarity).

**R1.C14:** Lines 598-599: this is a really important point that I'm glad was made (i.e., the marginal improvement might be not worth the effort). It seems to me that dealing with this issue is critical to making hybrid schemes more widely accepted. Is there any way we can determine a priori the marginal improvement (without having to build both models in parallel and then compare)? For example, the Mai et al. (2022) study in line 616 – would be good to comment if the demonstrated superiority was enough to justify the extra effort.
Yes, we agree that this is an important but tricky point. We have addressed this point in the last section, which is now titled "Interpretability, usability, and uptake of hybrid forecasts":
"One issue that is critical to making hybrid schemes more widely accepted is determining whether the improvement in forecast skill obtained by building a hybrid model is worth the extra effort. In other words, it can be difficult to determine *a priori* how much added value can be obtained without first developing the hybrid model and benchmarking the results against a more traditional approach. (…)
The benchmarking study of Mai et al. (2022) provided a detailed intercomparison of modelling approaches over the Great Lakes region (USA and Canada), suggesting that the effort related to ML is justifiable. However, this work was for retrospective simulation, rather than forecasting (for which there are more steps needed) and therefore it is still a jump to suggest that ML always provides improvements for prediction, particularly over seasonal to decadal horizons, for which studies are lacking. (…)
Implementing an operational hybrid framework for hydroclimatic forecasting often requires extensive time and expertise, given that two completely different types of models must be developed in parallel. These requirements would also likely require a shift in the expertise of the organisation as well as an upgrade in the computing architecture in the case of GPU-requiring hybrid and data-driven approaches."

**R1.C15:** Table 1: (a) Are any of these operational? (b) Any rationale for inclusion/exclusion of studies in this table? (c) Can you add another column that describes how the statistical and dynamial models are combined? (d) Regarding column headings, in the text, "data-driven" seems to be the most generic term (lines 25-26) but here the column header is "statistical" model (and elsewhere, "empirical" is used). Again, the importance of terminology in this paper. (e) Would this table become slightly easier to digest if it was first sorted by predictand type (i.e., streamflow vs. reservoir, etc) and then horizon? I'm not sure, but I think that predictand is a larger category (and what I would first be interested in), then horizon.
Table 1 in the original manuscript, which lists examples of hybrid models from the literature, is now Table 2 in the revised manuscript.
(a) We included some operational examples, such as those mentioned in our reply to **R1.C3**. We also revised the text to make it clearer that hybrid hydroclimate forecasting is a form of operational practice already.
(b) We sought to cover different types of dynamical and statistical models, different ranges, and different variables (as there are too many papers to include all). Hence, we have ensured there is a representative sample of all the different study types listed in Table 3.
(c) We included a new column in the new Table 2 and improved the new Table 1 (types of hybrid models).
(d) The column heading has been updated to "data-driven".
(e) We made the choice to sort by horizon, because we felt it was more important to emphasize the applicability of the method across a range of horizons, but we appreciate the idea.
Thank you for the useful suggestions!

**R1.C16:** Some acronyms that are not defined anywhere: RCP8.5, FV3GFS (this is just the name of the atmospheric model?), PREVAH (also a model name?)
The definitions of these acronyms have all been added to Table 2: Representative Concentration Pathway 8.5 (high-emissions warming scenario); Finite-Volume Cubed-Sphere Global Forecast System (global atmospheric model); and Precipitation-Runoff-Evapo-transpiration Hydrotope Model. We also checked the text to make sure we had not accidentally missed any other acronyms, and added these to the table.

**R1.C17:** Table 3: (a) Shouldn't "coupled" be included here also, since it is discussed in the text. (b) I find it interesting that Lee et al. (2002) is a primary reference for two of the options (serial and parallel) – given that it is now 20 years ago. Is that because it was such a foundational paper? Either way, can a more recent reference also be provided? As a corollary comment: It would be nice to have a discussion in the text of when these approaches were first tried (what was the foundational paper) on hydroclimate variables.

We have substantially updated the new Table 1 (former Table 3) to address these points, provide a more comprehensive overview of the different types of hybrid structure that exist, and to better clarify and describe the approaches in the main text. Please see revised table in response to **R1.C1**.

**R1.C18:** Figure 1: A few comments/questions on this graphic: (a) Please explain if the coloration of the boxes has any meaning. (b) Aren't large-scale predictors etc. also inputs to the hybrid forecasting scheme (not just dynamical predictors) – in other words, the straightforward left-to-right is not actually quite so straightforward? (c) Bottom middle: shouldn't it be "hydroclimate model" rather than "hydrological model" to be more general?

Thank you for helping us make the figure more intuitive. We have revised it and replaced the expression "statistical or machine learning" with "data-driven" for consistency with the revised manuscript.

(a) The colour of the boxes indicates the broad type of prediction scheme and serves to help the reader see how the top two schemes (rows) are combined in the third scheme (bottom row, reflecting hybrid prediction); we have clarified the figure caption accordingly (please see revised caption below).

(b) Yes, large-scale predictors can also be used as inputs, but would likely be issued from dynamical predictions or dynamical reanalyses (e.g., using large scale principal components to identify predictors) in the case of a hybrid forecast (although some observations might be employed too). We have updated the figure caption to make it clear that multiple different types of combinations are possible.

(c) Yes, we agree that the bottom middle box would be better with the term "hydroclimate model" and have updated it accordingly; thank you for spotting this.

[Figure]

Figure 1. Defining hybrid hydroclimate forecasting and prediction. `Hydroclimate' refers to a range of variables defined in the text, including streamflow. The top row indicates traditional dynamical hydroclimate predictions (blue); middle row is data-driven (DD) predictions (yellow) and bottom row represents hybrid predictions (red), which combine dynamical and data-driven predictions. In the last row, three examples of hybrid structure are shown from top to bottom: (i) Statistical-dynamical (Stat-dyn), (ii) Serial, and (iii) Coupled, as described in Table 1. The figure provides simple examples, but other schemes are possible, including for example a mix of observations and predictions in the left column.

**R1.C19:** Figure 2: So, you obtain one value each for JJA, then take the max? Could be clarified in the caption text.

The maximum summer discharge is the largest of the 92 daily values in the June-July-August period. The caption has been revised to state this explicitly.

Thank you for this constructive review!

**Response to Reviewer 2**
**Anonymous Referee #2, 09 Nov 2022**

**Summary**

This paper reviews - indeed it defines - the burgeoning field of hybrid dynamical-statistical hydrometeorological forecasting. The paper is timely and I believe it to be of wide interest to readers of HESS (and very likely beyond). I generally like to balance positive and negative feedback in reviews, but it was very difficult for me to find any suggestions to improve in this paper. It is skillfully organised, placing a very wide range of studies in sensible categories and highlighting specific themes with more detailed discussions of some papers. I didn't think there were really any major gaps in the literature and ideas they presented. The paper is also brilliantly written, with concise, lucid sentences making it an easy read - I believe even for non-experts. In short, in my view this review does everything a review should do: summarises the literature comprehensively, shapes the literature sensible themes, makes an argument - in this case the paper is essentially arguing for the recognition of hybrid forecasting as a distinct field (or at least a subfield within hydrometeorological forecasting) - and makes clear recommendations on the future direction of hybrid forecasting. I congratulate the authors on a remarkable review paper, one that I believe deserves to be widely cited.

Reply. We are most grateful to the Reviewer for this kind assessment of our work! The Reviewer's comments are copy-pasted below verbatim in black font, and our replies are in blue font. We label the comments in the following manner: "R2.C1" indicates Reviewer 2, Comment 1.

**Specific comments**

**R2.C1:** L33 "We do not provide a prescriptive definition of hybrid forecasting as it exists along a continuum, which may include a wide range of modeling and 'big data' type Earth Observation (EO) datasets" Fair enough - a sensible choice.

We are glad the Reviewer agrees with this choice!

**R2.C2:** L156 "ML models are also employed during the dynamical climate model simulations to correct model biases" I suspect the use of 'ML' to describe Bayesian techniques like Schepen and bias-correction methods like Meyer may be a bit unusual to many. Suggest the broader term 'statistical models' or 'data driven models' (consistent with the definition given in the introduction) to encompass all these.

We have updated this to "data-driven models".

"Data-driven models are employed during the dynamical climate model simulations to correct model biases"

**R2.C3:** L156 "The use of ML..." same issue with this paragraph - I would say that neither Bennett et al. nor McInerney et al. really qualify as ML - they are error models, which I think in general usage don't get lumped in with ML. These distinctions may well be arbitrary, but I'd suggest if the authors want to broaden the common use of ML to include a wide range statistical models that this be defined up front somewhere (in the way the authors have done with 'data-driven').

We agree and have updated this paragraph to "data-driven models" also.

"Data-driven approaches can also be applied directly to post-process the hydrological forecasts."

**R2.C4:** L453 "4 Key challenges and opportunities of hybrid forecasting" I guess I would add to the topics covered in this section the effective use of probabilistic forecasts in decision making. One of the major efforts in hybrid forecasting systems has been to achieve reliable predictive distributions; but it's not yet clear that this effort will necessarily result in better decisions. It's likely that automated decision systems/optimisation will be the means to take advantage of reliability in ensemble distributions. In my view this still requires considerable research -

existing methods of optimsation do not necessary take advantage of this property. But I also understand that this may be outside the scope of what the authors wish to address - the paper is really comprehensive in the areas it does choose to address, so they may feel they cannot do this area justice (even if they agree that it is worth discussing). I will leave it to the authors to decide whether this is worth including in their paper.

We entirely agree with the Reviewer that the development of probabilistic forecasts and their subsequent uptake in decision making (and potential for improving decisions) is an important topic to address. However, this topic is relevant for all ensemble and probabilistic systems, hybrid or not; therefore, in the end (after some debate) we decided not to include a discussion of this point in the revised manuscript.

**R2.C5:** L456 "ML models include the requirement for large datasets (previously discussed)" This review presents the availability of large datasets for ML as a strength of ML - which it of course is - but it presents few of the difficulties associated with using these datasets for prediction, for example some of the 'curse(s) of dimensionality' described by Altman & Krzywinski (2018). ML models are still subject to some of these issues - though I realise canvassing these is not the main aim of the paper. Whether these matters are best discussed in this paper is a subjective judgment: I am happy to defer to the authors on this point.

Thank you for this nice suggestion. We have included some explanation of the difficulties associated with the use of large datasets for hybrid prediction, based on this reference.

"As hybrid forecasts and predictions rely on data-driven models, they inevitably inherit some of the limitations of these techniques. Frequently-cited limitations of ML models include the requirement for large datasets and issues associated with the `curse of dimensionality', namely data sparsity (i.e. when there are too few data points relative to the number of dimensions), multicollinearity of the variables, multiple testing (leading to an increased number of false positives), and overfitting (Altman and Krzywinski, 2018)." (…)

We have also emphasised some of these difficulties may be less applicable in seasonal to decadal forecasting:

"While some of the difficulties associated with large sample sizes apply less for seasonal to decadal hybrid forecasting, where the sample sizes can be much smaller (often near 100 values) than the sample sizes for shorter ranges (thousands or more), the small sample sizes present a challenge for model training. Thus, a range of different challenges may apply depending on the forecasting horizon and data required."

**R2.C6:** L465 "data-driven models were once thought to be unable to accurately predict values outside the range of the training" I'm not sure this is really true (or if it is, I haven't been exposed to it) - would be good to provide a reference in support of this statement. There is a long history of statistical extrapolation - not least in extreme value theory or design engineering - for exactly these purposes.

It is interesting that there seem to be different opinions on the question of data extrapolation by data-driven models. After reviewing the literature on this point, we find that it is difficult to find any reliable comparisons, and therefore have revised this paragraph accordingly.

"One important challenge of hybrid models is the need to produce physically-plausible or explainable forecasts in unseen extreme conditions such as severe floods, droughts, intense heatwaves and tropical storms. This is particularly important as new weather records are being set in different parts of the world, and models must produce credible predictions under extreme forcing conditions. Although it has sometimes been suggested that data-driven models might be less suited to extrapolation to out-of-sample conditions than physics-based models due to the lack of physical understanding (e.g. Reichstein et al., 2019), recent work tackled the question of whether modern LSTMs could predict events larger than those seen in the training data for a particular catchment. The authors found that the LSTM could predict 'unseen' streamflow extremes, and did this better than the physics-based models that were used in the study (Frame et al., 2022a). It is now increasingly recognised that one of the advantages of data-driven models is their flexibility, allowing them to find unexpected patterns in the data. Thus, there are emerging synergies between data-driven and physics-based approaches, since the former can enhance the performance of the latter, e.g. by learning the parameterizations required for the physical models from large datasets or analysing the patterns of error from the physical models."

**R2.C7:** L487 "Explainability is sometimes useful to help develop trust in model predictions" this is a very interesting point - in my experience forecasting agencies frequently engage in this kind of story-telling, both for internal and external communications, so this is probably an important box to tick for the widespread adoption of hybrid forecasting systems. I'm not suggesting any change here, but I guess I also feel this kind of narrative building can be antithetical to the effective use of (usually carefully constructed) probability distributions that come out of hybrid forecasting systems.

We agree this is an interesting point for discussion too. We have updated this paragraph to reflect the different perspectives on the use of storytelling/narratives versus the use of probabilistic forecasts.

"Explainability is sometimes useful to help develop trust in model predictions. Forecasting agencies frequently engage in a form of story-telling, both for internal and external communications. One reason for providing explainable predictions is that when the forecasts evolve for a given variable, such as spring runoff, users often wish to understand why (i.e. what has changed in the predictors or other factors to explain the change in the predictions). One way to achieve explainability is by providing storylines or narratives around the hybrid forecasts which demonstrate the geophysical credibility of the results. Differentiable modelling can also provide diverse physical variable outputs, trained or untrained, which help develop a narrative (Feng et al. 2022). Fleming et al. (2021) showed how hydroclimatic storylines can be produced for clients to make the forecast interpretable in terms of understandable geophysical processes. They used pragmatic methods such as `popular votes' for the candidate predictors cast by a genetic algorithm. The approach revealed how the values of predictors such as antecedent flow and snow water equivalent could help explain the ensemble mean predicted volume. However, there are also limitations to such approaches. Although narratives may help with stakeholder acceptance of hybrid forecasting systems, they can also form a constraint on the forecasting approach, by enforcing consistency of a given prediction method."

**R2.C8:** L536 "For low flows skill may currently extend up to 20 days, but this is mostly due to the quality of the information on initial conditions and the memory effect of catchment storage" this statement may be true specifically for the study by Fundel et al. 2013, but it is phrased more generally. It is quite possible to get forecast skill of streamflow well beyond twenty days - even with simple ESP methods - (depending on catchment, time of year, etc.) so I think the authors should avoid a statement that posits a general limit on the prediction of streamflow of 20 days. Please reword this so that it is clear that this finding was specific to Fundel et al.

We have reworded this sentence so it is clear the finding is specific to Fundel et al., and that skill can be obtained beyond 20 days in other cases. Thank you.

"Low flows may have skill up to 20 days in the case of Fundel et al. (2013) and even longer in other cases, especially with good information on initial conditions and/or the memory effect of catchment storage."

**R2.C9:** Fig 4: As you've used 'prediction' generically in the vertical axis label ('Prediction skill') - implying (correctly in my view) that all the models in this plot produce predictions - I suggest changing the label "Subseasonal to seasonal predictions" to the more specific "Subseasonal to seasonal forecasts" and the label "Climate predictions" to "Multi-year climate forecasts".

These are excellent points, thank you very much! We have revised the figure accordingly, as shown below. The caption was also edited to make this point even more explicit.

[Figure]

Figure 4. Hybrid models could be a promising route for seamlessly linking initialized predictions from seasonal and decadal forecasts to scenario-based projections across timescales.(…)

**Typos etc.**

L50 "While conceptual hydrological models..." suggest a paragraph break before 'While'
Done.

L71 Suggest paragraph break before 'Historically...'
Done; thank you.

L83 "to understand to which" typo - delete second 'to'
The sentence has been rewritten for clarity.

"Research is required to understand the hydro-climatological conditions to which new ML and DL models are able to extrapolate from the training set, and their performance as they are extrapolated in space."

**References**

Altman N, Krzywinski M. 2018. The curse(s) of dimensionality. Nature Methods 15: 399-400. DOI: 10.1038/s41592-018-0019-x.

Thank you for the reference, which has been added to the revised manuscript.

Thank you for the helpful review!